

**Approaches to radar reflectivity bias correction to improve**
**rainfall estimation in Korea**
**C.-H. You[1], M.-Y. Kang[2], D.-I. Lee[1,2], and J.-T., Lee[2]**
[1] {Atmospheric Environmental Research Institute, Pukyong National University, Busan,
South Korea }
[2] {Department of Environmental Atmospheric Sciences, Pukyong National University,
Busan, South Korea}
Correspondence to: D.-I. Lee (leedi@pknu.ac.kr)
**Abstract**
Three methods for determining the reflectivity bias of single polarization radar using dual
polarization radar reflectivity and disdrometer data (i.e., the equidistance line, overlapping
area, and disdrometer methods) are proposed and evaluated for two low-pressure rainfall
events that occurred over the Korean Peninsula on 25 August 2014 and 8 September 2012.
Single polarization radar reflectivity was underestimated by more than 12 dB and 7 dB in the
two rain events, respectively. All methods improved the accuracy of rainfall estimation,
except for one case where DSDs were not observed, as the precipitation system did not pass
through the disdrometer location. The use of these bias correction methods reduced the RMSE
by as much as 50%. Overall, the most accurate rainfall estimates were obtained using the
overlapping area method to correct radar reflectivity. A combination of all three methods
would produce more accurate rainfall estimates, provided optimal values are determined for
the domain size for the overlapping area method, the sample number threshold for the
equidistance line method, and the reflectivity threshold for the disdrometer method.
**1    Introduction**
Radar is a useful remote sensing instrument for measuring rainfall amount, due to its
relatively high resolution in both space and time. Rainfall rate is not measured directly, but
must be derived from radar reflectivity. This derivation of radar rainfall is based on the



relationship between reflectivity (Z) and rainfall rate (R), known as the Z–R relation (R(Z)).
Experimentally measured drop size distributions (DSDs) have been used extensively to obtain
both radar reflectivity and rainfall rate (Compos and Zawadzki, 2000; Jang et al., 2004; You
et al., 2004). There does not be existed a unique R(Z), since DSDs can vary between storms
and even within a single storm (Battan 1973; You et al., 2010).
However, radar rainfall estimation is complicated by a number of uncertainties including
hardware calibration, partial beam filling, rain attenuation, brightband, and non-weather
echoes (Wilson and Brandes, 1979; Austin, 1987). The correction of bias in Z caused by
hardware calibration error is difficult to achieve using single polarimetric radar (SPOL) alone.
Polarimetric radar (DPOL) provides a new method for the absolute calibration of reflectivity,
which has been a longstanding problem with single polarization radar data. The method is
based on the assumptions that Z, differential reflectivity ($Z_{DR}$), and specific differential phase
($K_{DP}$) are independent of each other, and that Z can be estimated from $Z_{DR}$ and $K_{DP}$, which are
insensitive to radar miscalibration (Gorgucci et al., 1992, 1999; Goddard et al., 1994;
Scarchilli et al., 1996; Vivekanandan et al., 1999).
The Korea Meteorological Administration (KMA) is in the process of replacing Doppler
radars with S-band DPOLs (to be completed by 2019), and Ministry of Land, Infrastructure
and Transport (MoLIT) has installed four S-band DPOLs for operational use since 2009. Until
the DPOL installation is complete, it is necessary to use a combination of SPOLs and DPOLs
to produce rainfall mosaics covering the whole Korean Peninsula. To obtain more accurate
mosaicked radar rainfall, SPOL reflectivity should be corrected using the reflectivity of
DPOLs and other instruments such as disdrometer. Accurate SPOL reflectivity is also
required for climatological analysis using radar rainfall.
This paper discusses three methods for reducing errors in SPOL reflectivity using DPOL and
DSD measurements. In Section 2, the dataset used for the analysis is introduced, and three
approaches to correcting SPOL reflectivity are described, along with methods for bias
correction of DPOL reflectivity and $Z_{DR}$, and for validation. In Section 3, the results obtained
using the three correction methods are compared with gauge measurements. Finally, we
summarize the results and provide conclusions in Section 4.





## 2 Data and methodology

### 2.1 Gauge, disdrometer, and radar datasets

Rainfall data from rain gauges operated by the KMA were used to evaluate the accuracy of radar rainfall. Rain gauges located between 5 and 134 km from the radar were included in the analysis. Figure 1 shows the location of all instruments used in this study. The PARSIVEL (PARticle SIze VELocity) disdrometer was installed ~9 km from PSN. PARSIVEL is a laser-optic system that measures 32 channels from 0.062 to 24.5 mm (for detailed specifications, see Loffler-Mang and Joss, 2000).

Data were regarded as unreliable and removed from the analysis in the case that any of the following conditions were met: 1 min rain rate was less than 0.1 mm h$^{-1}$; total number concentration from all channels was less than 10; drop numbers were recorded only in the lower 10 channels (1.187 mm for PARSIVEL); or drop numbers were recorded only in the lower 5 channels (0.562 mm for PARSIVEL) (You et al., 2015).

Radar data were recorded at PSN and BSL, which were installed and are operated by KMA and MoLIT, respectively. The transmitted peak power of BSL is 750 kW, the beam width is 0.95 °, the frequency is 2.791 GHz, and the antenna is 1085 m above sea level. The polarimetric variables are estimated with a gate size of 0.125 km. The scan strategy consists of six elevation angles with a 2.5 min update interval. The transmitted peak power of PSN is 800 kW, the beam width is 1.0 degrees, and the antenna is 547 m above sea level. The reflectivity is estimated with a gate size of 0.25 km. The PSN scan strategy consists of 13 elevation angles with a 10 min update interval. Radar variables at an elevation angle of 0.5 (1.8) degrees were extracted from the BSL (PSN) data every 10 mins, to match the time interval for this study. Non-meteorological targets were removed from the PSN data using the texture and vertical gradient of reflectivity, as proposed by Zhang et al. (2004). Polarimetric variables were subjected to quality control using a threshold of 15 degrees for the standard deviation of the differential phase shift (You et al., 2014).

### 2.2 Methodology for bias correction of PSN reflectivity

To calculate the reflectivity bias of PSN, which is single polarization radar, three approaches were used: the equidistance line method, the overlapping area method, and the disdrometer method. The first approach is to compare the reflectivities along the line that is equidistant



between the two radars. To determine this line for the two radars, the effective radius was set
to 100 km and the distance between the two radars and the azimuthal angle pointing from
BSL to PSN were calculated using their latitude and longitude values. The start and end
azimuthal angles for comparison of reflectivity were then calculated as follows:
$AZ_{st} = \beta - a\cos(0.5 \times dr/rc)$      (1)
$AZ_{end} = \beta - a\cos(0.5 \times dr/rc) + 2 \times a\cos(0.5 \times dr/rc),$     (2)
where $AZ_{st}$ and $AZ_{end}$ are the start and end azimuthal angles for the comparison, respectively;
$\beta$ is an azimuthal angle which is the angle between north and the bearing from BSL points to
PSNand $rc$ and $dr$ are the effective radius and distance from BSL to PSN, respectively. The
distance between the two radars is 76.9 km, and the start and end azimuthal angles of DPOL
(SPOL) are 79 (35) and 213 (261) degrees, respectively (Fig. 2).
To compare the reflectivity observed of targets at similar heights from both radars, the beam
height was calculated assuming a standard atmospheric beam propagation (Rinehart, 2010), as
follows:
$H = \sqrt{r^2 + (R'+H_0)^2 + 2r(R'+H_0)\sin\phi} - R',$   (3)
where $r$ is the slant range from the radar, $\Phi$ is the elevation angle of the radar beam, $H_0$ is the
height of the radar antenna above sea level, and R' = (4/3)R, where R is the Earth's radius
(6,371 km). The radar antenna heights of SPOL and DPOL are 547 and 1085 m, respectively.
Figure 3 shows the beam height of PSN and BSL at the equidistance line. EL1 to EL6 show
the elevation angles from smallest to largest. The smallest difference in beam height between
the two radars is 149 m, which was obtained using the fourth elevation angle of PSN and the
third elevation angle of BSL.
In the second approach, the overlappingping area for the two radars was calculated by
matching the coordinates. The polar coordinate of two radars was converted to a Cartesian
coordinate with a spatial resolution of 1 km. The overlapping area was then determined by
multiplying the distances between the two radars in the east–west and north–south directions.
Figure 4 shows a schematic diagram of the overlapping area for the two radars. The extracted
domain of PSN and BSL for the comparison is 158 × 136 km.





The third and final approach is to use DSD observations from the PARSIVEL disdrometer.
The reflectivity was calculated from the DSD measurements at 1 min resolution, and averaged
over 10 mins to match the radar time resolution. Figure 5 shows a schematic of the procedure
used to match the radar and PARSIVEL data. The PARSIVEL disdrometer is located ~9 km
from the radar, at an azimuthal angle of 87 degrees. The radar reflectivity was averaged over a
domain of 13 gates × 3 degrees in azimuth, centered at the PARSIVEL location.

### 2.3   Z and $Z_{DR}$ bias correction for BSL

Before calculating reflectivity bias for PSN using BSL, reflectivity and $Z_{DR}$ must be corrected
for systematic bias. $Z_{DR}$ bias correction is important for the absolute calibration of the radar
using a self-consistency method. Gorgucci et al. (1999) proposed using a vertical pointing
scan of light rain, to take advantage of the nearly spherical shape of the raindrops as seen
from below. Ryzhkov et al (2005) used the elevation angle dependency of $Z_{DR}$ as an
alternative technique and concluded that the high variability of $Z_{DR}$ in rainfall prohibited the
method from achieving the required absolute calibration accuracy of 0.2 dB. They instead
proposed a method that utilizes the structural characteristics of the melting layer in stratiform
clouds and the dry aggregated snow present above the melting layer. $Z_{DR}$ measurements from
dry aggregated snow above the melting layer resulted in a mean S-band value of 0.2 dB and
an accuracy of 0.1–0.2 dB. Trabal et al. (2009) evaluated two methods using the intrinsic
properties of dry aggregated snow present above the melting layer and light rain
measurements close to the ground, and found that a $Z_{DR}$ calibration accuracy of 0.2 dB or
better was achieved using either method.
Vertical pointing data were not available in the present case, and the scan strategy, with six
elevation angles, was unable to detect the melting layer. Therefore, in this study, light rain
measurements close to the ground were used to calibrate $Z_{DR}$ and reflectivity using a self-
consistency method. Light rain was defined using a threshold of 20 dBZ $\leq$ Z $\leq$ 28 dBZ, as
proposed by Marks et al. (2011). The $Z_H$ bias was calculated following the method of
Ryzhkov et al. (2005), using a 9-gate moving average of $Z_{DR}$ to improve the accuracy.





## 2.4  Validation

The normalized error (NE), root-mean-square error (RMSE), and correlation coefficient (CC) between rainfall estimates and measurements from 121 gauges were calculated to measure the performance of each bias correction method. These quantities are defined as follows:

$$NE = \frac{\frac{1}{N}\sum_{i=1}^{N}\left|R_{R,i} - R_{G,i}\right|}{\overline{R_G}} \qquad (3)$$

$$RMSE = \left[\frac{1}{N}\sum_{i=1}^{N}(R_{R,i} - R_{G,i})^2\right]^{1/2} \qquad (4)$$

$$CC = \frac{\sum_{i=1}^{N}(R_{R,i} - \overline{R_R})(R_{G,i} - \overline{R_G})}{\left[\sum_{i=1}^{N}(R_{R,i} - \overline{R_R})^2\right]^{1/2}\left[\sum_{i=1}^{N}(R_{G,i} - \overline{R_G})^2\right]^{1/2}}, \qquad (5)$$

where N is the number of radar rainfall ($R_R$) and gauge rainfall ($R_G$) pairs, and $\overline{R_R}$ and $\overline{R_G}$ are the average hourly rain rates from radar and gauges, respectively. These quantities were calculated using 1 hour rainfall amounts from radar and gauge measurements at each point. The radar rainfall value at each point was obtained by averaging rainfall over a small area (1 km × 1°) centered on the corresponding rain gauge. The radar rainfall was calculated using the relation $Z = 200\,R^{1.6}$.

## 3  Results

The accuracy of rainfall estimation using corrected reflectivity was evaluated to measure the effectiveness of each method for calculating reflectivity bias. Two rainfall events were used, occurring on 25 August 2014 and 8 September 2012 (Table 1). The August and September events were caused by low pressure systems over southern and northern Korea, respectively.

### 3.1  Equidistance line method

Before estimating radar rainfall rates, reflectivity biases were calculated using each of the three methods. Figure 6 shows time series of the average reflectivity difference between PSN and BSL at the equidistance line and the number of samples used in each calculation, on 25





August 2014. The average difference over the entire time period was –7.85 dB, and the largest
difference was –12.46 dB. The number of samples used for each calculation was determined
using a beam height difference threshold of 0.1 km. The total number of the samples
satisfying the threshold along the equidistance line was 77. The number of samples was
generally above 40, but it was smaller than 40 at 1120 LST and after 1500 LST. Figure 7
shows the same information for 8 September 2012. The average reflectivity difference over
the entire time period was – 2.56 dB, and the largest difference was –6.77 dB. The number of
samples was less than 50 until 0310 LST, after which it increased to more than 60. This result
suggests that the precipitation system was not located over the equidistance line until 0310
LST.
Figure 8 shows the scatter plot of 1 hour radar rainfall, calculated using $Z = 200\ R^{1.6}$,
gauge rainfall, for 25 August 2014 and 8 September 2012. The RMSE, NE, and CC for
rainfall pairs on 25 August 2014 were improved from 65.7 to 32.6 mm, from 0.79 to 0.36, and
from 0.88 to 0.89, respectively. On 8 September 2012, the RMSE, NE, and CC changed from
30.0 to 22.5 mm, from 0.58 to 0.41, and from 0.81 to 0.78, respectively, by the use of bias
correction. In both cases, the use of corrected reflectivity for rainfall estimation resulted in
much better accuracy than did using raw reflectivity.
## 3.2   Overlapping area method
Figure 9 shows time series of the mean reflectivity differences between PSN and BSL in the
overlapping area, and the number of samples used in each calculation (black squares) on 25
August 2014. Bias values ranged from –11.7 to –8.3 dB over the period analyzed. The bias
was stable until 1440 LST, after which it fluctuated as the number of samples decreased.
Figure 10 shows the same information for 8 September 2012. Bias values ranged from –4.66
to 0.22 dB, and did not show fluctuations due to low sample numbers.
Figure 11 shows a scatter plot of 1 hour radar rainfall, calculated using $Z = 200\ R^{1.6}$,
gauge rainfall, for 25 August 2014 and 8 September 2012. The RMSE and NE of rainfall pairs
on 25 August 2014 were improved from 65.7 to 29.7 mm and from 0.79 to 0.31, respectively.
On 8 September 2012, RMSE and NE were improved from 30.0 to 21.8 mm and from 0.58 to
0.40, respectively, by the use of bias correction, while CC was unchanged at 0.81. Again, in
both cases the use of corrected reflectivity for rainfall estimation was found to improve the
accuracy compared with raw reflectivity.





### 3.3 Disdrometer method

Before using the disdrometer bias correction method to estimate rainfall rates, 10 min rain rates obtained directly from DSDs and from collocated gauges were compared. Figure 12 shows the time series of rain rate obtained by PARSIVEL (red circles) and collocated gauges (blue circles) on 25 August 2014. Daily total rainfall amounts for PARSIVEL and the gauges were 129.4 and 116.0 mm, respectively. The difference in the totals is only 13.4 mm, and the RMSE and CC between the 10 min time series were 0.52 mm h$^{-1}$ and 0.99, respectively. On 8 September 2012 (not shown), the difference between the total daily rainfall amounts was 0.7 mm and the RMSE and CC between the two 10 min series were 0.62 mm h$^{-1}$ and 0.96, respectively. It is concluded that DSDs were sufficiently reliable to use as a reference with which to calculate the radar bias.

Figure 13 shows time series of reflectivity obtained by radar (black circles) and by PARSIVEL (red circles), and the radar bias (blue circles), on 25 August 2014. The bias was more stable before 1200 LST than after 1400 LST. PARSIVEL reflectivity fell to zero from 1230 to 1340 LST because the precipitation system moved away from the PARSIVEL site. Because of this discontinuity, the bias can be considered to be reliable only until 1200 LST. Figure 14 shows time series of reflectivity obtained by radar (black circles) and by PARSIVEL (red circles), and the radar bias (blue circles), on 8 September 2012. On this occasion there was no reflectivity data from either PARSIVEL or radar until 0330 LST.

Figure 15 shows a scatter plot of hourly radar rainfall, calculated using Z = 200 R$^{1.6}$, and gauge rainfall, on 25 August 2014 and 8 September 2012. The RMSE and NE of rainfall pairs on 25 August 2014 were improved from 65.7 mm to 42.0 mm and from 0.79 to 0.40, respectively. On 8 September 2012, RMSE and NE decreased from 30.1 to 24.6 mm, and from 0.58 to 0.46, respectively, while CC decreased from 0.81 to 0.65. In both cases, using corrected rather than raw reflectivity for rainfall estimation improved accuracy as measured by RMSE and NE, but reduced accuracy as measured by CC.

### 3.4 Discussion

Figure 16 shows hourly rainfall RMSE from each of the different bias correction methods on 25 August 2014 and 8 September 2012. Red, black, green, and blue bars show the RMSE obtained using the uncorrected, equidistance line, overlapping area, and disdrometer methods, respectively. The disdrometer method produced the lowest RMSE before 1200 LST and the




highest RMSE after 1200 LST (Fig. 16a). This behavior can be attributed to the varying
stability of the reflectivity calculated by PARSIVEL (Fig. 13). The overlapping method is
more accurate than the equidistance line method for the entire time period, except at 1400
LST. All the bias correction methods performed better than the uncorrected method, except
for the period during which DSDs were unavailable. On 8 September 2012, the RMSE of the
overlapping area method was lower than that of the other methods for the entire period,
except at 0500 and 0600 LST (Fig. 16(b)). The disdrometer method produced lower RMSE at
0600 LST, when DSDs were available, and the equidistance line method was more accurate at
0500 LST, when the sample number was high (Fig. 13). Considering the entire period
covering both events, the overlapping area method showed the best performance, as measured
by RMSE. The accuracy of radar rainfall estimates could be improved by combining the three
approaches, using metrics such as DSD temporal stability and the number of samples
available for the equidistance line method to select the best method for a particular situation.
**4    Conclusions**
Three methods for determining the reflectivity bias of single polarization radar using dual
polarization radar reflectivity and disdrometer data were proposed and examined for two
rainfall events caused by low pressure over the Korean Peninsula on 25 August 2014 and 8
September 2012. Single polarization radar reflectivity was underestimated by more than 12
dB and 7 dB during the August and September events, respectively. All three methods
improved the accuracy of estimated rainfall, except during a period when DSDs were not
observed (as the precipitation system did not pass over the disdrometer location). The use of
these bias correction methods reduced rainfall RMSE by up to 50%. Overall, the accuracy of
rainfall estimation was highest when the overlapping area method was used to correct radar
reflectivity.
The reflectivity biases obtained using the disdrometer and equidistance line methods were
more temporally variable than those obtained using the overlapping area method. There were
several hours during which the disdrometer method was more accurate than the overlapping
area method. We suggest that combining the overlapping area method with the disdrometer
method, using threshold criteria such as the temporal stability of reflectivity and the number
of samples available would allow more accurate estimates of rainfall. However, optimum
values for the domain size for the overlapping area method, the sample number threshold for





the equidistance line method, and the reflectivity threshold for the disdrometer method should
be determined in order to combine the three methods most effectively.
**Acknowledgements**
The authors thank the Ministry of Land, Infrastructure, and Transport of the Korean
government and the Korean Meteorological Administration for providing radar data and AWS
(Automatic Weather System) gauge data. This research was funded by the Korea
Meteorological Industry Promotion Agency under Grant KMIPA 2015-1050. And this
research was partly funded by the Korea Meteorological Industry Promotion Agency under
Grant KMIPA 2015-1060





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





Table 1. Rainfall events used for the analysis.

| Date | Source | Period of analysis |
|---|---|---|
| 8 September 2012 | Low pressure | 0000 LST to 0600 LST |
| 25 August 2014 | Low pressure | 0900 LST to 1600 LST |



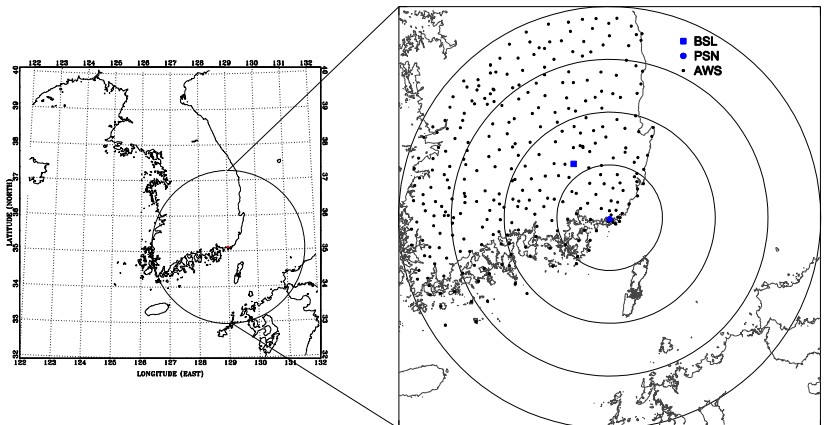

Figure 1. Location of the Bislsan radar (solid rectangle), the PARSIVEL disdrometer and
Gudeok radar (solid circle), and rain gauges (black dots) distributed within 240 km of radar
coverage. Circles indicate distance from the Gudeok radar, and are drawn at intervals of 60
km.



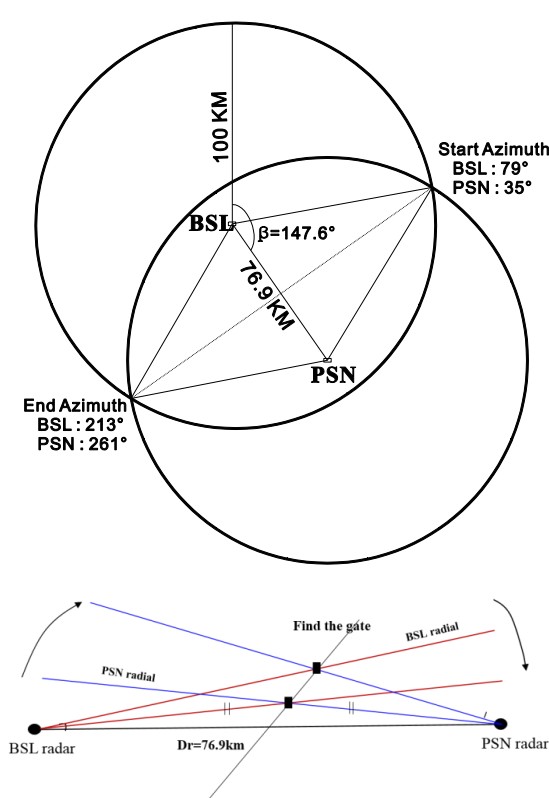

Figure 2. Schematic diagram showing the method used to calculate the line of equidistance
between two radars. The effective radius was set to 100 km and the distance between radars is
76.9 km. The azimuthal angle from BSL to PSN is 147.6 degrees. The start and end azimuthal
angles are 79 (35) and 213 (261) degrees for BSL (PSN), respectively.





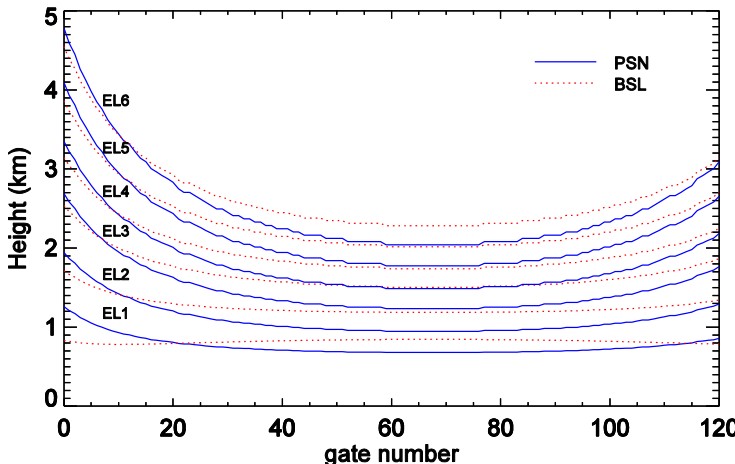

2    Figure 3. Beam height of PSN (blue solid lines) and BSL (red dotted lines) at the equidistance

3    line. EL1 to EL6 show the lowest, second, third, fourth, fifth, and sixth elevation angles,

4    respectively.







4    Figure 4. Schematic diagram of the overlapping area for BSL and PSN. The east–west and

5    north–south distances between the two radars are 42 km and 64 km, respectively.





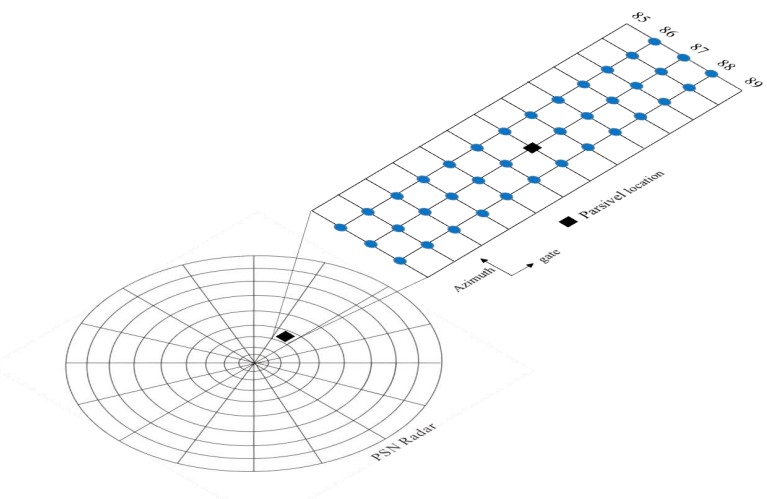

Figure 5. Schematic diagram showing matching of the radar gate and the PARSIVEL
disdrometer. PARSIVEL is located ~9 km from the radar, at an azimuthal angle of 87 degrees.
The radar reflectivity was averaged over a 3 km × 3° domain centered at the PARSIVEL
location.





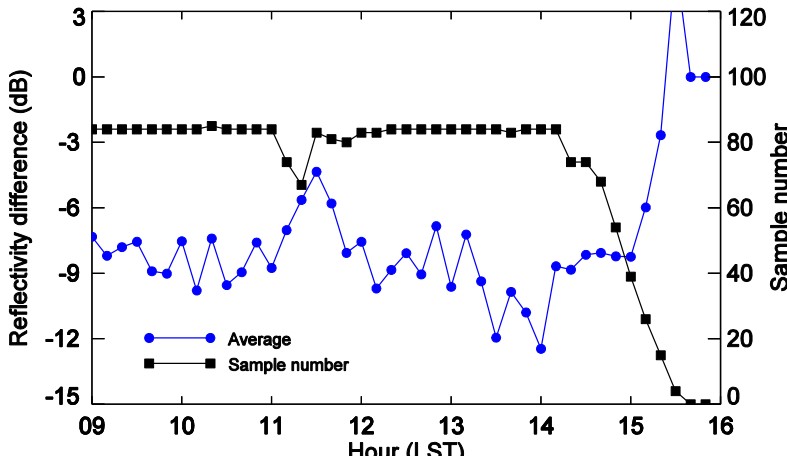

3    Figure 6. Time series of the average reflectivity difference between PSN and BSL at the

4    equidistance line (blue circles) and the number of samples used in each calculation (black

5    squares) on 25 August in 2014.





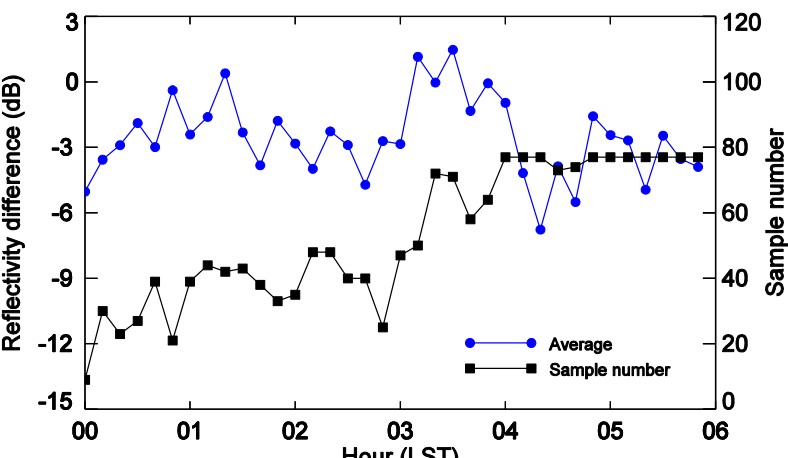

3    Figure 7. As for Fig. 6 but for 8 September 2012.





(a)                                     (b)

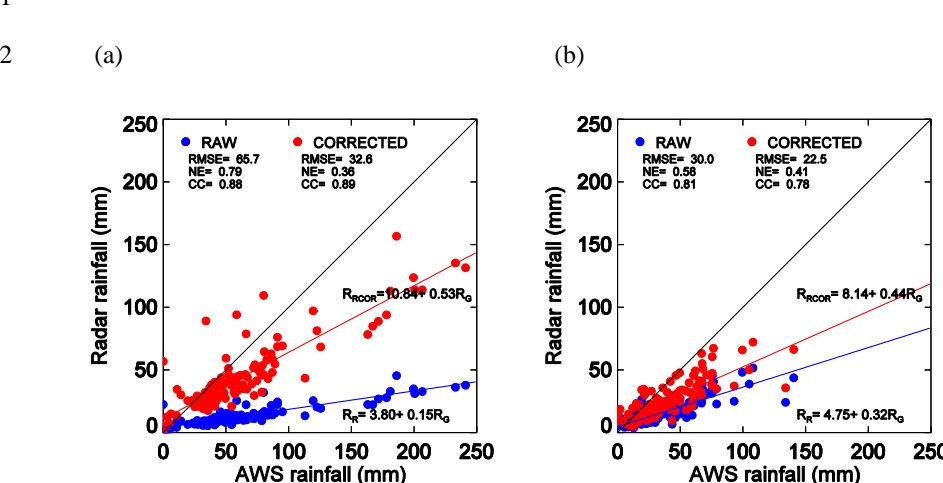

Figure 8. Scatter plot of hourly radar rainfall, calculated using $Z = 200 R^{1.6}$, and gauge rainfall,
for (a) 25 August 2014 and (b) 8 September 2012. Blue circles show the rainfall pairs
obtained using raw reflectivity and red circles show those obtained using reflectivity corrected
with the equidistance line method.



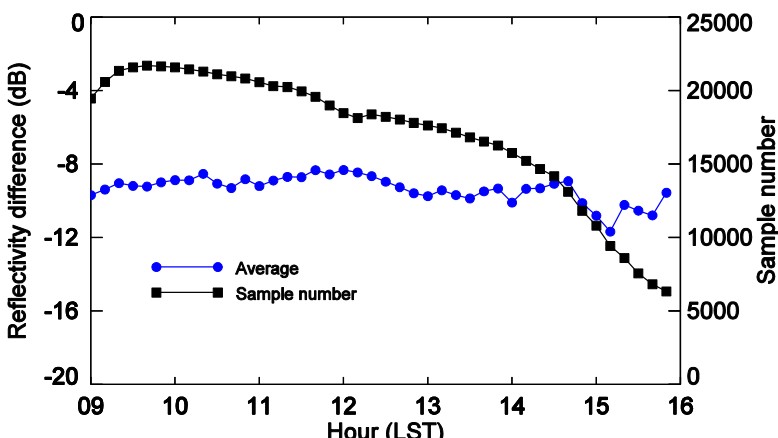

3    Figure 9. As for Fig. 6 but for the overlapping area method.



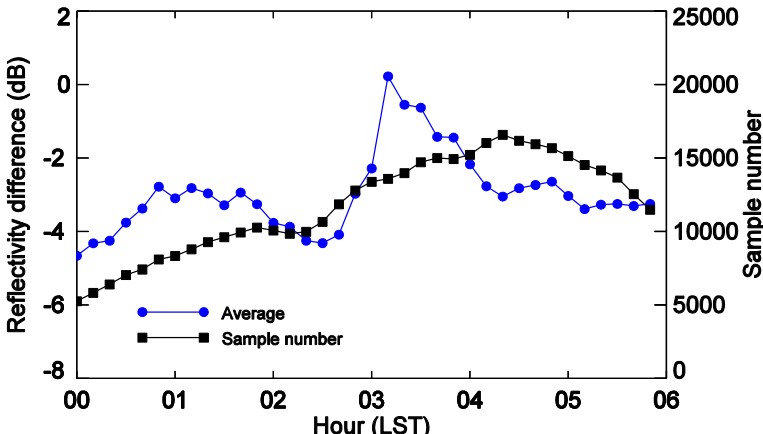

3    Figure 10. As for Fig. 7 but for the overlapping area method.





2      (a)                                 (b)

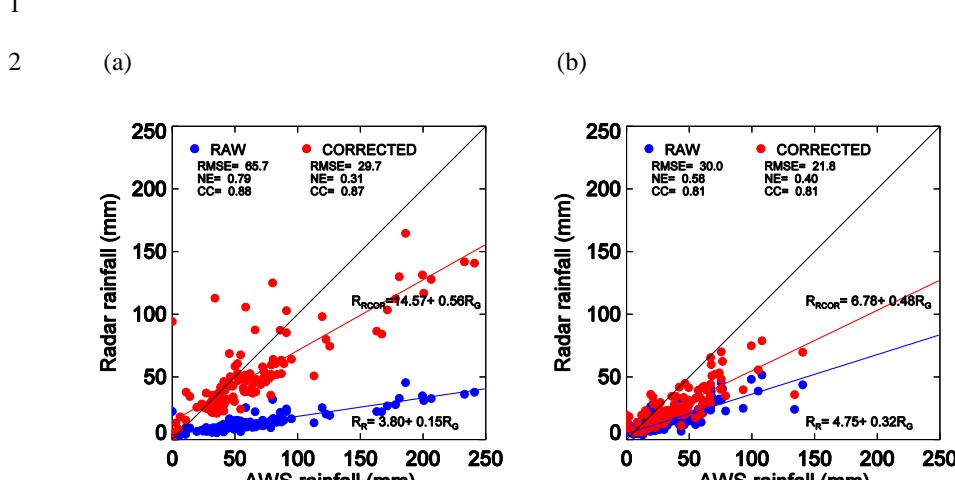

4    Figure 11. As for Fig. 8 but for the overlapping area method.



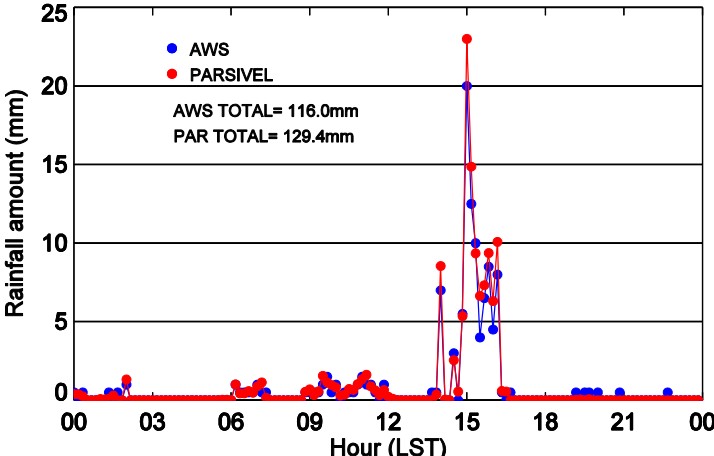

3 Figure 12. Time series of 10 min rainfall amount as obtained by PARSIVEL (red circles) and

4 collocated gauges (blue circles).





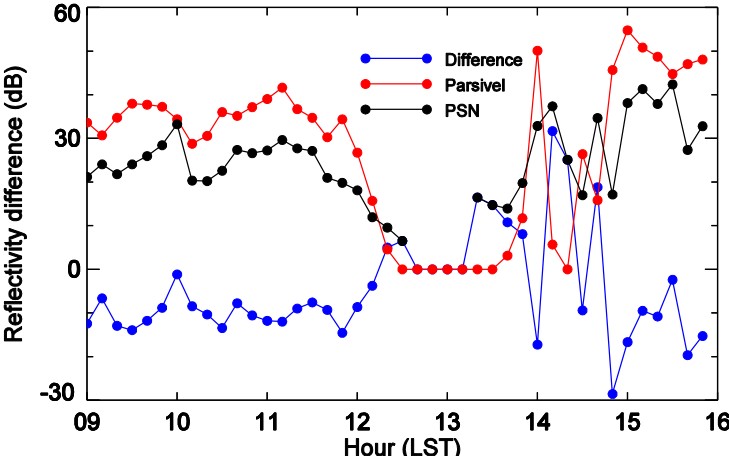

3    Figure 13. Time series of reflectivity obtained by radar (black circles) and by PARSIVEL (red

4    circles), and the radar bias (blue circles) on 25 August 2014.



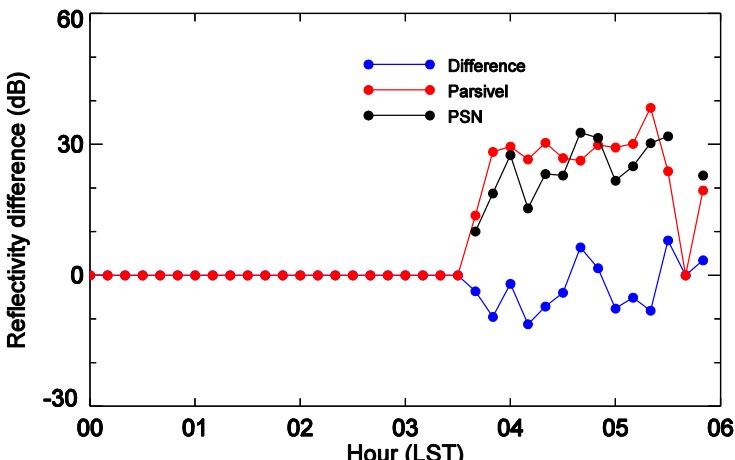

3    Figure 14. As for Fig. 13 but for 8 September 2012.





2      (a)                           (b)

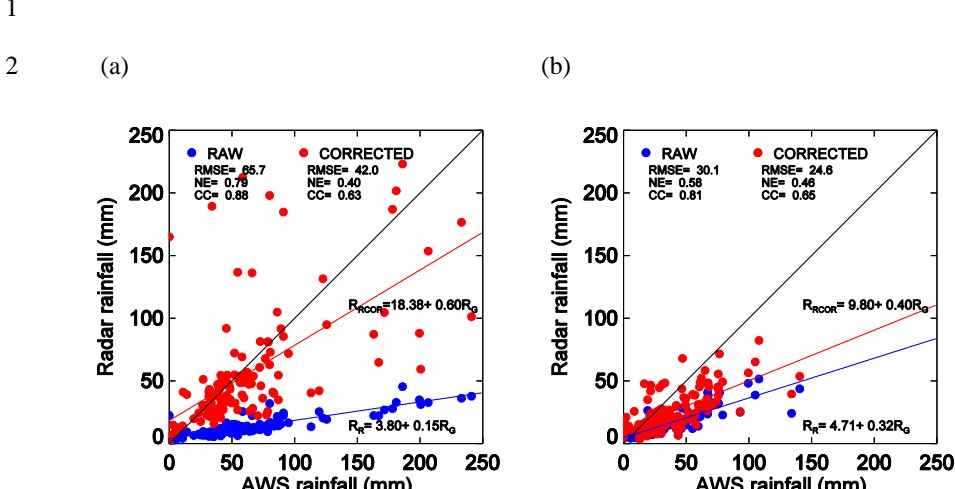

4      Figure 15. As for Fig. 8 but for the disdrometer method.





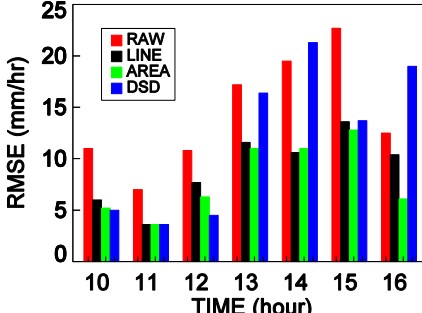

Figure 16. Hourly rainfall RMSE for different bias correction methods on 25 August 2014
(left) and 8 September 2012 (right). The bars with different colors show results obtained using
the raw data, equidistance line method, overlapping area method, and disdrometer method,
respectively.

