# Peer review of "Approaches to radar reflectivity bias correction to improve"

_Atmospheric Measurement Techniques, 2015_

## Referee Comment (RC1) · Anonymous Referee #1 · 17 Feb 2016

The following is a review of amt-2015-392

Primary comment: It is understood that the authors desire the best calibration possible for a single polarizarion radar (SPOL) using dual-polarimetric radar (DPOL) as a guide. In the opinion of this reviewer, the authors do not describe the 3 calibration methodologies clearly, and do not describe the meteorological events and data clearly. The authors draw conclusions as to which methodology is best based on statistics generated from a fixed ZR relationship. This approach is very misleading as a fixed ZR relationship may only be valid for a limited time/area within the event. Figure 16 shows noticeable differences in the RMSE statistics between the two events, yet the authors do not describe why. I suspect the differences are due to the underlying meteorological differences between the two events analyzed. By analyzing other events, conclusions drawn by the same methodology may be completely different depending on the degree

of validity of the fixed ZR.

If the authors wish to revise this paper, I respectfully suggest that the focus be on the three calibration techniques, and not on matching rain gauge data. There is not enough description of the calibration methods to fully understand what the authors are actually doing. In section 2.3, the authors state that the light rain threshold of 20 dBZ <= Z <= 28 dBZ was used in Ryzhkov et al 2005 and Marks et al 2011 for self-consistency calibration – this is not correct. By using such light rain, the Kdp values will not be high enough for reliable self-consistency results. No description of the self-consistency equation is provided - what equation was used, and how was it derived? There is not an adequate description of the Zdr calibration. There is no explanation as to how reflectivity values between SPOL and DPOL are actually compared.

Figure captions are not well described (i.e. fig 9 and 10), and lead to confusion of the reader. Large fluctuations in the reflectivity differences are not described (other than by decreased sample numbers).

Authors are basing conclusions on rain gauge rates, yet little-to-no information is provided on the gauge network. What type of rain gauges? Given that rain gauges are an accumulation instrument, how are rain rates computed.....via interpolation as in Wang et al 2008, or some other method?

Correcting radar calibration will improve comparisons with ground "truth" instruments. This is nothing new to the research community. If the authors wish to move forward with this paper, I respectfully suggest that the emphasis be placed on the actual calibration techniques, and not focus on potentially misleading results from a fixed ZR relationship.

Suggested disposition: Major revisions or rejection in current form.

———————————————————

---

## Author Comment (AC1) · 22 Feb 2016

Response to review

At first, thank you very much for referee's effort in reviewing our paper even your busy time. We revised the manuscript titled "Approached to radar reflectivity bias correction to improve rainfall estimation in Korea" that was submitted to Atmospheric Measurement Techniques. The manuscript has been revised as suggested by reviewer and we also corrected some mistakes. We would appreciate any feedback on the revisions.

Response to review by Anonymous referee #1

Primary comment 1. It is understood that the authors desire the best calibration possible for a single polarization radar (SPOL) using dual-polarimetric radar (DPOL) as a guide. In the opinion of this reviewer, the authors do not describe the 3 calibration methodologies clearly

Author's Response:

Thank you for your comment. We added some descriptions to make more clearly for reader's easier understanding regarding with three calibration methodologies. First, we added the following sentences from line 12 to 14 on page 6 in the manuscript for the equidistance method. "Therefore, the reflectivity bias of PSN was calculated by averaging the difference of reflectivity along with the equidistance line observed from fourth elevation angle of PSN and third one of BSL." And we also added "Equidistance line" in Fig. 2 for better understanding and removed the bottom figure were removed for the simplicity. Secondly, we added some sentences for better understanding of overlapping method from 19 to 22 line on page 6 in the manuscript as follows; "The distance between two radars in east-west and north-south direction are 42 km and 64 km, respectively. The reflectivity observed from both radars at the pixels designated at the overlapping area as shown by blue rectangle in right panel of Fig. 4 were compared to calculate the ZH bias of PSN." Finally, we described the following sentences for DSD method from 29 to 31 line on page 6 in the manuscript as follows; "The difference of reflectivity observed from PSN and PARSIVEL were calculated and was then taken as a ZH bias."

2. Authors do not describe the meteorological events and data clearly.

Author's Response:

Thank you for your kind comment. We added the following sentence for describing the data used in this study more clearly from line 27 to 30 on page 3 in the manuscript. "The quality controlled ZH, ZDR, KDP measured from BSL were used to calibrate ZDR and ZH of BSL. The ZH measured from PSN were then corrected by using calibrated ZH of BSL using self-consistency method and ZH measured by PARSIVEL. The gage rainfall data were used to assess the performance of three ZH bias correction methods

for PSN which is SPOL." We also added two Figures (Figures 6 and 7) to explain the meteorological events clearly in the manuscript. We explained the events used for the study from 22 line on page 7 to 20 line on page 8 as follows; "Figure 6 shows the time series of ZH observed from BSL radar on 8 September in 2012 and 25 August in 2014. The precipitation within radar coverage on 8 September in 2012 was occurred by low pressure with the front located at northern part of Korea. The core of the precipitation systems was elongated from south to north and moved to eastward. The maximum reflectivity of the core was more than 45 dBZ and caused rainfall at the western part of radar center at 0300 LST (Fig. 6(a)), became more organized shape at the eastern part of radar center at 0400 LST (Fig. 6(c)), and moved to eastward and located out of land at 0500 LST (Fig. 6(e)) on 8 September in 2012. The precipitation system on 25 August in 2014 was caused by the low pressure located at southern part of Korea. The two strong rainfall within the radar coverage were located at south-western part of radar center with distance between 120 km and 150 km and southern part of radar center with distance between 30 km and 90 km, respectively at 1200 LST on 25 August in 2014 (Fig. 6(b)). The two convective cells moved to eastward, their strength were intensified and the area of rainfall was wider at 1300 LST (Fig. 6(d)). The two systems moved to eastward continuously, were merged together at 1400 LST (Fig. 6(f)). Figure 7 shows the time series of hourly rainfall and daily accumulation measured by a gage which recorded highest daily rainfall within radar coverage on 8 September in 2012 and 25 August in 2014. The highest daily accumulated rainfall was recorded from North Changwon (ID 255) and Geumjeong (ID 939) on each day, respectively. The daily accumulation of ID 255 was 150 mm, the maximum hourly rainfall was around 40 mm, and the duration of the rainfall was 7 hours (Fig. 7.(a)). The daily accumulation of ID 939 was around 270 mm, the maximum hourly rainfall was more than 100 mm h-1. The rainfall amount for 3 hours (1000 LST, 1400 LST, and 1500 LST) were mainly contributed to the total rainfall accumulation on 25 August in 2014 (Fig. 7(b))." We also changed Figure number in the manuscript, accordingly.

3. The authors draw conclusions as to which methodology is best based on statistics

generated from a fixed ZR relationship. This approach is very misleading as a fixed ZR relationship may only be valid for a limited time/area within the event.

Author's Response:

Thank you for your comment. As reviewer's comment, there are many uncertainties of radar rainfall estimation. The variability of ZR relation is one of main source of these uncertainties. The ZR relations are different from storm to storm, precipitation types, different climatology, and so on. In this study, we would like to propose the methods for correcting ZH of SPOL which have trouble in calibration using well calibrated ZH from DPOL and PARSIVEL. And we also would like to assess the performance of three methods. One of way to understand their performance is to use the radar rainfall estimation. That is why we calculated radar rainfall estimation using corrected ZH measured by each method. For better understanding of three methods, we added the results of validations obtained by another ZR relation, Z=300R1.4 which is widely used for NEXRAD in the manuscript. We also tested another rainfall system caused by the front with indirect effect of Typhoon on 23 August in 2012 using same methods. The results are summarized in Table 1(Please refer to Figure 1). The results were similar to those of manuscript. Therefore, we assume that the radar rainfall estimation would be used for examining the performance of the three methods.

4. Figure 16 shows noticeable differences in the RMSE statistics between the two events, yet the authors do not describe why. I suspect the differences are due to the underlying meteorological differences between the two events analyzed. By analyzing other events, conclusions drawn by the same methodology may be completely different depending on the degree of validity of the fixed ZR.

Author's Response:

Thank you for your comment. As reviewer's comment, we added the following sentences from line 13 to 16 on page 11 in the manuscript. "It would be caused by the difference of total rainfall amount between two rainfall systems. The maximum total rainfall amount for both cases were around 250 mm for 25 August and 150 mm for 8 September 2012. Another reason of the fluctuation would be the difference of radar hardware calibration error for PSN between two events. "

5. If the authors wish to revise this paper, I respectfully suggest that the focus be on the three calibration techniques, and not on matching rain gauge data. There is not enough description of the calibration methods to fully understand what the authors are actually doing.

Author's Response:

Thank you for your kind comment. For better understanding, we changed the Section in 2. Data and methodology as follows; 2.2 Z and ZDR bias correction of BSL, 2.3 Methodology for bias correction of PSN reflectivity. We also added the sentences as mentioned in the answer to the number 1 reviewer's comment in the manuscript.

6. In section 2.3, the authors state that the light rain threshold of 20 dBZ<= Z <=28 dBZ was used in Ryzhkov et al 2005 and Marks et al 2011 for self-consistency calibration – this is not correct. By using such light rain, the Kdp values will not be high enough for reliable self-consistency results. No description of the self-consistency equation is provided – what equation was used, and how was it derived ?

Author's Response:

Thank you for your comment and I am sorry to make reviewer confused. We used the threshold of 20 dBZ<= Z <=28 dBZ for ZDR bias calculation. Marks et al. (2011) used the threshold to categorize the light rain. We followed the method proposed by Ryzhkov et al. (2005) for ZH bias calculation of BSL. We also added the self-consistency method used for this study in the Sect. 2.2 of the manuscript.

7. There is not an adequate description of the Zdr calibration.

Author's Response:

Thank you for your comment. We added the following description of ZDR calibration in the manuscript." The assumption of ZDR is close to zero in case of the small rain drop like drizzle was chosen for this study. The ZDR observed from BSL having with reflectivity in the range of 20 dBZ to 28 dBZ for given time period were averaged. Then the averaged ZDR was taken as a ZDR bias."

8. There is no explanation as to how reflectivity values between SPOL and DPOL are actually compared.

Author's Response: Thank you for your comments. For the equidistance method, we extracted the reflectivity from PSN and BSL at the equidistance line as shown in Fig. 2. The averaged difference reflectivity extracted from PSN and BSL were taken as a ZH bias. For the overlapping method, we extracted the reflectivity from PSN and BSL at the overlapping area as shown in Fig. 4. The averaged difference reflectivity extracted from both radar were taken as a ZH bias. Before the comparison, the reflectivity measured from BSL were calibrated by self-consistency method. For DSD method, PSN reflectivity was averaged over a domain 13 gates by 3 degrees centered at the PAR-SIVEL location. Then the averaged difference reflectivity of PSN and PARSIVEL were taken as a ZH bias. We added above mentioned sentences in the manuscript.

9. Figure captions are not well described (i. e. fig 9 and 10), and lead to confusion of the reader. Large fluctuations in the reflectivity differences are not described (other than by decreased sample numbers).

Author's Response: Thank you for your comment. We modified the captions of Fig. 9 and 10 and also described the following sentences from line 17 to line 22 on page 9. "lower bias values were occurred from 0300 LST to 0400 LST. The fluctuation also would be caused by the sudden change of microphysical characteristics of rainfall pass through the overlapping area for both radars. It would reduce the accuracy of ZH of BSL corrected by self-consistency. The radar rainfall estimation was done by using observed and corrected ZH as an input of Z-R relations." And we also modified Figures and 16 for simplicity and correction for the legend.

10. Authors are basing conclusions on rain gague rates, yet little-to-no information is provided on the gauge network. What type of rain gauges? Given that rain gauges are an accumulation instrument, how are rain rates computed.. via interpolation as in Wang et al 2008, or some other method?

Author's Response:

Thank you for your comment and I am sorry for confusion. We added the following gage information from line 5 to 6 on page 7. "The rain gages were 0.5 mm tipping-bucket type. Time resolution of gages is 1 min and data quality control was done by KMA." We did not convert gage rainfall into rain rate but we converted radar rain rate to rainfall amount. We used the accumulated rainfall amount calculated from radar and gage for the validation. We also modified "hourly rainfall" to "total accumulated rainfall amount" in the manuscript, accordingly.

11. Correcting radar calibration will improve comparisons with ground "truth" instruments. This is nothing new to the research community. If the authors wish to move forward with this paper, I respectfully suggest that the emphasis be placed on the actual calibration techniques, and not focus on potentially misleading results from a fixed ZR relationship.

Author's Response:

Thank you very much for your really kind comment. As mentioned the answer to the reviewer's comment number 3, we would like to to propose the methods for correcting ZH of SPOL which have trouble in calibration using well calibrated ZH from DPOL and PARSIVEL. And we also would like to assess the performance of three methods. One of way to understand their performance is to use the radar rainfall estimation. That is why we calculated radar rainfall estimation using corrected ZH measured by each method. Please understand our proposal.

\*\*\* Thank you very much again for your deep review and it will be of much help for better our manuscript quality.\*\*\*

Please also note the supplement to this comment:
http://www.atmos-meas-tech-discuss.net/amt-2015-392/amt-2015-392-AC1-supplement.pdf
* * *
[Figure]

Table 1. The RMSE, NE, and CC obtained from radar ($Z=200R^{1.6}$) and gage rainfall on 23 August in 2012

| Statistics | Raw ZH | Corrected ZH | | |
| --- | --- | --- | --- | --- |
| | | Equidistance | Overlapping | DSD |
| RMSE | 62.6 | 45.1 | 27.8 | 49.4 |
| NE | 0.6 | 0.39 | 0.25 | 0.43 |
| CC | 0.85 | 0.82 | 0.83 | 0.76 |

**Fig. 1.**

**Supplement:**

[revised manuscript text omitted]

The quality controlled $Z_H$, $Z_{DR}$, $K_{DP}$ measured from BSL were used to calibrate $Z_{DR}$ and $Z_H$ of BSL. The $Z_H$ measured from PSN were then corrected by using calibrated $Z_H$ of BSL using self-consistency method and $Z_H$ measured by PARSIVEL. The gage rainfall data were used to assess the performance of three $Z_H$ bias correction methods for PSN which is SPOL.

**2.2 Z and $Z_{DR}$ bias correction for BSL**

Before calculating reflectivity bias for PSN using BSL, reflectivity and $Z_{DR}$ must be corrected for systematic bias. $Z_{DR}$ bias correction is important for the absolute calibration of the radar using a self-consistency method. Gorgucci et al. (1999) proposed using a vertical pointing scan of light rain, to take advantage of the nearly spherical shape of the raindrops as seen from below. Ryzhkov et al (2005) used the elevation angle dependency of $Z_{DR}$ as an alternative technique and concluded that the high variability of $Z_{DR}$ in rainfall prohibited the method from achieving the required absolute calibration accuracy of 0.2 dB. They instead proposed a method that utilizes the structural characteristics of the melting layer in stratiform clouds and the dry aggregated snow present above the melting layer. $Z_{DR}$ measurements from dry aggregated snow above the melting layer resulted in a mean S-band value of 0.2 dB and an accuracy of 0.1–0.2 dB. Trabal et al. (2009) evaluated two methods using the intrinsic properties of dry aggregated snow present above the melting layer and light rain measurements close to the ground, and found that a $Z_{DR}$ calibration accuracy of 0.2 dB or better was achieved using either method.

Vertical pointing data were not available in the present case, and the scan strategy, with six elevation angles, was unable to detect the melting layer. Therefore, in this study, light rain measurements close to the ground were used to calibrate $Z_{DR}$. Light rain was defined using a threshold of $20\ dBZ \leq Z \leq 28\ dBZ$, as proposed by Marks et al. (2011). The assumption of $Z_{DR}$ is close to zero in case of the small rain drop like drizzle was chosen for this study. The $Z_{DR}$ observed from BSL having with reflectivity in the range of 20 dBZ to 28 dBZ for given time period were averaged. Then the averaged $Z_{DR}$ was taken as a $Z_{DR}$ bias.

The $Z_H$ bias was calculated by self-consistency method using a 9-gate moving average of bias corrected $Z_{DR}$ in the range of 0.2 dB to 3.0 dB to improve the accuracy. This method depends on the notion that $Z_H$, $Z_{DR}$, and $K_{DP}$ are independent in rain, and that $Z_H$ can be estimated from $Z_{DR}$ and $K_{DP}$. The difference between the computed and observed values of $Z_H$ is referred to as the Z bias. Following the method of Ryzhkov et al. (2005), the entire spatial and temporal domain was divided into 1 dB intervals of $Z_H$ between Zmin (30 dBZ) and Zmax (50 dBZ), and the $K_{DP}(Z_H)$ and $Z_{DR}(Z_H)$ within each interval were calculated. The $Z_H$ bias is then determined by matching the integrals:

$$I_1 = \sum_{Z_{min}}^{Z_{max}} K_{DP}(Z) n(Z) \Delta Z \,, \tag{1}$$

$$I_2 = \sum_{Z_{min}}^{Z_{max}} 10^{0.1Z_m} f(Z_{DR})n(Z)\Delta Z ,$$  (2)

The function of f($Z_{DR}$) in Eq. (2) can be well approximated by a fourth-order polynomial fit for certain range of $Z_{DR}$ (Gourley et al., 2009) like Eq. (3).

$$f(Z_{DR}) = 10^{-5}(a_0 + a_1 Z_{DR} + a_2 Z_{DR}^2 + a_3 Z_{DR}^3) ,$$  (3)

The estimated $Z_H$ bias is determined from Vivekanandan et al. (2003) by

$$Z_H bias(dB) = 10 \log(\frac{I_2}{I_1}) ,$$  (4)

If the radar is well calibrated, $Z_H$ bias should be equal to 0. The coefficients of f($Z_{DR}$) were calculated by T-matrix scattering method using long period DSD data and are 4.26, -4.67, 2.67, and -0.54, respectively.

**2.3  Methodology for bias correction of PSN reflectivity**

To calculate the reflectivity bias of PSN, which is single polarization radar, three approaches were used: the equidistance line method, the overlapping area method, and the disdrometer method. The first approach is to compare the reflectivities along the line that is equidistant between the two radars. To determine this line for the two radars, the effective radius was set to 100 km and the distance between the two radars and the azimuthal angle pointing from BSL

to PSN were calculated using their latitude and longitude values. The start and end azimuthal angles for comparison of reflectivity were then calculated as follows:

$$AZ_{st} = \beta - a\cos(0.5 \times dr / rc)$$  (1)

$$AZ_{end} = \beta - a\cos(0.5 \times dr / rc) + 2 \times a\cos(0.5 \times dr / rc) ,$$  (2)

where $AZ_{st}$ and $AZ_{end}$ are the start and end azimuthal angles for the comparison, respectively; $\beta$

is an azimuthal angle which is the angle between north and the bearing from BSL points to

PSNand $rc$ and $dr$ are the effective radius and distance from BSL to PSN, respectively. The distance between the two radars is 76.9 km, and the start and end azimuthal angles of BSL (PSN)

are 79 (35) and 213 (261) degrees, respectively (Fig. 2).

To compare the reflectivity observed of targets at similar heights from both radars, the beam height was calculated assuming a standard atmospheric beam propagation (Rinehart, 2010), as follows:

$H = \sqrt{r^2 + (R'+H_0)^2 + 2r(R'+H_0)\sin\phi} - R'$,   (3)

where $r$ is the slant range from the radar, $\Phi$ is the elevation angle of the radar beam, $H_0$ is the height of the radar antenna above sea level, and R' = (4/3)R, where R is the Earth's radius (6,371 km). The radar antenna heights of PSN and BSL are 547 and 1085 m, respectively.

Figure 3 shows the beam height of PSN with blue solid line and BSL at the equidistance line (blue dashed line as shown in Fig. 2). EL1 to EL6 show the elevation angles from smallest to largest. The smallest difference in beam height between the two radars is 149 m, which was obtained using the fourth elevation angle of PSN and the third elevation angle of BSL.

Therefore, the reflectivity bias of PSN was calculated by averaging the difference of reflectivity along with the equidistance line observed from fourth elevation angle of PSN and third one of

BSL.

In the second approach, the overlapping area for the two radars was calculated by matching the coordinates. The polar coordinate of two radars was converted to a Cartesian coordinate with a spatial resolution of 1 km. The overlapping area was then determined by considering the distances between the two radars in the east–west and north–south directions. Figure 4 shows a schematic diagram of the overlapping area for the two radars. The distance between two radars in east-west and north-south direction are 42 km and 64 km, respectively. The reflectivity observed from both radars at the pixels designated at the overlapping area as shown by blue rectangle in right panel of Fig. 4 were compared to calculate the $Z_H$ bias of PSN. The extracted domain of PSN and BSL for the comparison is $158 \times 136$ km.

The third and final approach is to use DSD observations from the PARSIVEL disdrometer. The reflectivity was calculated from the DSD measurements at 1 min resolution, and averaged over

10 mins to match the radar time resolution. Figure 5 shows a schematic of the procedure used to match the radar and PARSIVEL data. The PARSIVEL disdrometer is located ~9 km from the radar, at an azimuthal angle of 87 degrees. The radar reflectivity was averaged over a domain of 13 gates $\times$ 3 degrees in azimuth, centered at the PARSIVEL location. The difference of reflectivity observed from PSN and PARSIVEL were calculated and was then taken as a $Z_H$

bias.

**2.4  Validation**

The normalized error (NE), root-mean-square error (RMSE), and correlation coefficient (CC) between rainfall estimates and measurements from 121 gauges were calculated to measure the performance of each bias correction method. The rain gages were 0.5 mm tipping-bucket type. Time resolution of gages is 1 min and data quality control was done by KMA. These quantities are defined as follows:

$$NE = \frac{\frac{1}{N}\sum_{i=1}^{N}\left|R_{R,i} - R_{G,i}\right|}{\overline{R_G}} \tag{3}$$

$$RMSE = \left[\frac{1}{N}\sum_{i=1}^{N}(R_{R,i} - R_{G,i})^2\right]^{1/2} \tag{4}$$

$$CC = \frac{\sum_{i=1}^{N}(R_{R,i} - \overline{R_R})(R_{G,i} - \overline{R_G})}{\left[\sum_{i=1}^{N}(R_{R,i} - \overline{R_R})^2\right]^{1/2}\left[\sum_{i=1}^{N}(R_{G,i} - \overline{R_G})^2\right]^{1/2}}, \tag{5}$$

where N is the number of radar rainfall ($R_R$) and gauge rainfall ($R_G$) pairs, and $\overline{R_R}$ and $\overline{R_G}$ are the average hourly rain rates from radar and gauges, respectively. These quantities were calculated using total accumulated rainfall amounts for analyzed time period from radar and gauge measurements at each point. The radar rainfall value at each point was obtained by averaging rainfall over a small area (1 km × 1°) centered on the corresponding rain gauge. The radar rainfall was calculated using the relation $Z = 200 R^{1.6}$ and $Z = 300 R^{1.4}$.

**3  Results**

The accuracy of rainfall estimation using corrected reflectivity was evaluated to measure the effectiveness of each method for calculating reflectivity bias. Two rainfall events were used, occurring on 25 August 2014 and 8 September 2012 (Table 1). The August and September events were caused by low pressure systems over southern and northern Korea, respectively.

Figure 6 shows the time series of $Z_H$ observed from BSL radar on 8 September in 2012 and 25 August in 2014. The precipitation within radar coverage on 8 September in 2012 was occurred by low pressure with the front located at northern part of Korea. The core of the precipitation systems was elongated from south to north and moved to eastward. The maximum reflectivity of the core was more than 45 dBZ and caused rainfall at the western part of radar center at 0300

LST (Fig. 6(a)), became more organized shape at the eastern part of radar center at 0400 LST

(Fig. 6(c)), and moved to eastward and located out of land at 0500 LST (Fig. 6(e)) on 8

September in 2012.  The precipitation system on 25 August in 2014 was caused by the low pressure located at southern part of Korea. The two strong rainfall within the radar coverage were located at south-western part of radar center with distance between 120 km and 150 km and southern part of radar center with distance between 30 km and 90 km, respectively at 1200

LST on 25 August in 2014 (Fig. 6(b)). The two convective cells moved to eastward, their strength were intensified and the area of rainfall was wider at 1300 LST (Fig. 6(d)). The two systems moved to eastward continuously, were merged together at 1400 LST (Fig. 6(f)).

Figure 7 shows the time series of hourly rainfall and daily accumulation measured by a gage which recorded highest daily rainfall within radar coverage on 8 September in 2012 and 25

August in 2014. The highest daily accumulated rainfall was recorded from North Changwon (ID 255) and Geumjeong (ID 939) on each day, respectively. The daily accumulation of ID 255

was 150 mm, the maximum hourly rainfall was around 40 mm, and the duration of the rainfall was 7 hours (Fig. 7.(a)). The daily accumulation of ID 939 was around 270 mm, the maximum hourly rainfall was more than 100 mm h$^{-1}$. The rainfall amount for 3 hours (1000 LST, 1400

LST, and 1500 LST) were mainly contributed to the total rainfall accumulation on 25 August in 2014 (Fig. 7(b)).

**3.1  Equidistance line method**

Before estimating radar rainfall rates, reflectivity biases were calculated using each of the three methods. Figure 8 shows time series of the average reflectivity difference between PSN and

BSL at the equidistance line and the number of samples used in each calculation, on 25 August

2014. The average difference over the entire time period was −7.85 dB, and the largest difference was −12.46 dB. The number of samples used for each calculation was determined using a beam height difference threshold of 0.1 km. The number of samples was generally above 60, but it was smaller than 60 after 1450 LST. Figure 9 shows the same information for

8 September 2012. The average reflectivity difference over the entire time period was − 2.56

dB, and the largest difference was −6.77 dB. The number of samples was less than 50 until 0310

LST, after which it increased to more than 50. This result suggests that the precipitation system observed from both BSL and PSN radar was not located enough over the equidistance line to get a reliable comparison until 0310 LST.

Figure 10 shows the scatter plot of total accumulated radar rainfall amount for analyzed time period, calculated using $Z = 200 R^{1.6}$ and $Z=300R^{1.4}$ and gauge rainfall, for 25 August 2014 and

8 September 2012. The RMSE, NE, and CC of rainfall pairs for $Z = 200 R^{1.6}$ ($Z=300R^{1.4}$) on

25 August 2014 were improved from 65.7 (66.1) to 32.6 (27.0) mm, from 0.79 (0.81) to 0.36

(0.31), and from 0.88 (0.87) to 0.89 (0.88), respectively. On 8 September 2012, the RMSE, NE, and CC for $Z = 200 R^{1.6}$ ($Z=300R^{1.4}$) changed from 30.0 (28.5) to 22.5 (20.0) mm, from 0.58

(0.56) to 0.41 (0.36), and from 0.81 (0.8) to 0.78 (0.76), respectively, by the use of bias correction. In both cases, the use of corrected reflectivity for rainfall estimation resulted in much better accuracy than did using raw reflectivity.

**3.2   Overlapping area method**

Figure 11 shows time series of the mean reflectivity differences between PSN and BSL in the overlapping area, and the number of samples used for calculation of $Z_H$ bias on 25 August 2014.

Bias values ranged from –11.7 to –8.3 dB over the period analyzed. The bias was stable until

1440 LST, after which it fluctuated as the number of samples decreased. Figure 12 shows the same information for 8 September 2012. Bias values ranged from –4.66 to 0.22 dB, and lower bias values were occurred from 0300 LST to 0400 LST. The fluctuation also would be caused by the sudden change of microphysical characteristics of rainfall pass through the overlapping area for both radars. It would reduce the accuracy of $Z_H$ of BSL corrected by self-consistency.

The radar rainfall estimation was done by using observed and corrected $Z_H$ as an input of Z-R

relations.

Figure 13 shows a scatter plot of total accumulated radar rainfall amount for entire analyzed time period, calculated using $Z = 200 R^{1.6}$ and $Z=300R^{1.4}$ and gauge rainfall, for 25 August

2014 and 8 September 2012. The RMSE and NE of rainfall pairs for $Z = 200 R^{1.6}$ ($Z=300R^{1.4}$)

on 25 August 2014 were improved from 65.7 (66.1) to 29.7 (25.8) mm and from 0.79 (0.81) to

0.31 (0.28), respectively.  On 8 September 2012, RMSE and NE for $Z = 200 R^{1.6}$ ($Z=300R^{1.4}$)

were improved from 30.0 (28.5) to 21.8 (19.1) mm and from 0.58 (0.56) to 0.40 (0.34), respectively, by the use of bias correction, while CC for $Z=200R^{1.6}$ was unchanged at 0.81 and that of $Z=300R^{1.4}$ were changed 0.8 to 0.79. Again, in both cases the use of corrected reflectivity for rainfall estimation was found to improve the accuracy compared with raw reflectivity.

**3.3 Disdrometer method**

Before using the disdrometer bias correction method to estimate rainfall rates, 10 min rain rates obtained directly from DSDs and from collocated gauges were compared. Figure 14 shows the time series of rain rate obtained by PARSIVEL and collocated gauges on 25 August 2014. Daily total rainfall amounts for PARSIVEL and the gauges were 129.4 and 116.0 mm, respectively. The difference in the totals is only 13.4 mm, and the RMSE and CC between the 10 min time series were 0.52 mm h$^{-1}$ and 0.99, respectively. On 8 September 2012 (not shown), the difference between the total daily rainfall amounts was 0.7 mm and the RMSE and CC between the two 10 min series were 0.62 mm h$^{-1}$ and 0.96, respectively. It is concluded that DSDs were sufficiently reliable to use as a reference with which to calculate the radar bias.

Figure 15 shows time series of reflectivity obtained by radar and by PARSIVEL, and the radar bias, on 25 August 2014. The bias was more stable before 1200 LST than after 1400 LST. PARSIVEL reflectivity fell to zero from 1230 to 1340 LST because the precipitation system moved away from the PARSIVEL site. Because of this discontinuity, the bias can be considered to be reliable only until 1200 LST. The bias values ranged from -13.4 to -3.1 dB until 1200 LST. Figure 16 shows time series of reflectivity obtained by radar and by PARSIVEL, and the radar bias, on 8 September 2012. On this occasion there was no reflectivity data from either PARSIVEL or radar until 0330 LST. The bias values were distributed from -14.3 to 12.7dB.

Figure 17 shows a scatter plot of total accumulated radar rainfall amount for the entire time period, calculated using Z = 200 R$^{1.6}$ and Z=300R$^{1.4}$ and gauge rainfall, on 25 August 2014 and 8 September 2012. The RMSE and NE of rainfall pairs for Z = 200 R$^{1.6}$ (Z=300R$^{1.4}$) on 25 August 2014 were improved from 65.7 (66.1) mm to 42.0 (61.4) mm and from 0.79 (0.81) to 0.40 (0.53), respectively. On 8 September 2012, RMSE and NE for Z = 200 R$^{1.6}$ (Z=300R$^{1.4}$) decreased from 30.1 (28.6) to 24.6 (23.9) mm, and from 0.58 (0.56) to 0.46 (0.44), respectively, while CC for Z = 200 R$^{1.6}$ (Z=300R$^{1.4}$) decreased from 0.81 (0.8) to 0.65 (0.59). In both cases, using corrected rather than raw reflectivity for rainfall estimation improved accuracy as measured by RMSE and NE, but reduced accuracy as measured by CC.

**3.4 Discussion**

[revised manuscript text omitted]

location.

(a)            (b)

[Figure]

(c)            (d)

(e)            (f)

Figure 6. Time series of horizontal reflectivity (ZH) at 0.5 elevation angle observed from BSL

(a) 0400 LT, (c) 0500 LT, (e) 0600 LT on 8 September in 2012, (b) 1200 LT, (d) 1300 LT, (f)

1400 LT on 25 August in 2014.

(a)

[Figure]

(b)

[Figure]

Figure 7. Time series of 1 hour rainfall (bar) and daily accumulated (red line) measured from a gage which recorded highest daily rainfall within radar coverage at (a) North Changwon (ID

255) on 8 September in 2012 and (b) Geumjeong (ID 939) on 25 August in 2014.

[Figure]

Figure 8. Time series of the average reflectivity difference between PSN and BSL at the equidistance line (blue circles) and the number of samples used in each calculation (black squares) on 25 August in 2014.

[Figure]

Figure 9. As for Fig. 8 but for 8 September 2012.

1   (a)              (b)

[Figure]

3   (c)              (d)

5 Figure 10. Scatter plot of total accumulated rainfall for analyzed time period calculated by gage

6 and radar using (a and b) $Z = 200\ R^{1.6}$ and (c and d) $Z = 300\ R^{1.4}$ for 25 August 2014 and 8

7 September 2012, respectively. Blue circles show the rainfall pairs obtained using raw

8 reflectivity and red circles show those obtained using reflectivity corrected with the

9 equidistance line method.

[Figure]

2 Figure 11. As for Fig. 8 but for the overlapping area method.

[Figure]

Figure 12. As for Fig. 9 but for the overlapping area method.

(a)                                    (b)

[Figure]

(c)                                    (d)

Figure 13. As for Fig. 10 but for the overlapping area method.

[Figure]

Figure 14. Time series of 10 min rainfall amount as obtained by PARSIVEL (red circles) and collocated gauges (blue circles).

[Figure]

Figure 15. Time series of reflectivity obtained by PARSIVEL (red circles), and the radar bias (blue circles) on 25 August 2014.

[Figure]

Figure 16. As for Fig. 15 but for 8 September 2012.

(a)                                    (b)

[Figure]

(c)                                    (d)

Figure 17. As for Fig. 10 but for the disdrometer method.

(a)                                                     (b)

[Figure]

Figure 18. Accumulated rainfall RMSE calculated from radar and gage for different bias correction methods on (a) 25 August 2014 and (b) 8 September 2012. The bars with different colors show results obtained using the raw data, equidistance line method, overlapping area method, and disdrometer method, respectively.

---

## Referee Comment (RC2) · Anonymous Referee #2 · 4 Mar 2016

A. Major comments This paper discusses radar reflectivity bias correction to improve rainfall estimation in Korea. The paper content is relevant for its publication. However, for its publication, it needs some revisions as the followings;

1. Introduction (1) Line 28: rainfall rate -→ areal rainfall rate (2) Line 29: This derivation -→ this estimation (3) There does not be existed a unique R(Z),–→There is not existed a unique R(Z), (4) Put the values of ranges in figure 1 with interval of 60km.  2.2 Methodology (1) Line 12: at similar heights–→ at the almost same height (?) (2) The reflectivity was calculated from the DSD measurements at 1 min resolution,-→ The reflectivity was calculated from the DSD at 1 min resolution

3.1 Equidistance line method (1) Concerning Fig.6, it shows average reflectivity difference between PSN and BSL. Which radar shows higher reflectivity? What is the

reason on the dominant peak of the average value? It should be explained. (2) comparing the average reflectivity differences between PSN and BSL in Fig. 6 and 7, most of the average values in Fig. 7 is less than 0 dB. However, most of the average values in Fig. 6 is higher than 0 dB. It should be explained in detail.

3.2 Overlapping area method (1) Concerning Fig.10, it shows average reflectivity difference between PSN and BSL. Which radar shows higher reflectivity? What is the reason on the dominant peak of the average value around 03 LST? It should be explained.

3.3 Disdrometer method (1) What are the daily rainfall amounts from the gauges and Pasivel on 8 Sept. 2012? (2) explain the reason on the unstable behavior of reflectivity from Parsivel in Fig. 13. Is it reasonable to use Parsivel data as a reference in this study? Why is it sufficiently reliable to us as a reference in spite of the unstable behavior. * 4. Conclusions The authors should include comparison of three methods using some statistical parameters.

B. Reviewer's recommendation This paper shows radar reflectivity bias correction using three methods to get more accurate rainfall from single polarization radar. The approaches and results are considered as reasonable. Therefore, this paper is recommended for its publication with corrections as suggested in the major comments.

---

## Author Comment (AC2) · 7 Mar 2016

Response to review At first, thank you very much for referee's effort in reviewing our paper even your busy time. We revised the manuscript titled "Approached to radar reflectivity bias correction to improve rainfall estimation in Korea" that was submitted to Atmospheric Measurement Techniques. The manuscript has been revised as suggested by reviewer and we also corrected some mistakes. We would appreciate any feedback on the revisions.

Response to review by Anonymous referee 2 A. Major comments This paper discusses radar reflectivity bias correction to improve rainfall estimation in Korea. The paper content is relevant for its publication. However, for its publication, it needs some revisions as the followings;

[Figure]

1. 1. Introduction Line 28: rainfall rate → areal rainfall rate

Author's Response:

Thank you for your comment. We changed "rainfall rate" into "areal rainfall rate" at line 28 on page 1 in the revised manuscript.

2. 1. Introduction Line 29: This derivation -> this estimation

Author's Response:

Thank you for your comment. We changed "this derivation" to "this estimation" at line 29 on page 1 in the revised manuscript.

3. 1. Introduction, There does not be existed a unique R(Z) -> There is not existed a unique R(Z)

Author's Response:

Thank you for your comment. We changed the sentence accordingly at line 4 on page 2 in the revised manuscript as reviewer's suggestion.

4. Put the values of ranges in figure 1 with interval of 60 km.

Author's Response:

Thank you for your comment. We added the values of range with interval of 60 km on Figure 1 in the revised paper.

5. 2.2 Methodology Line 12: at similar heights -> at the almost same height(?)

Author's Response:

Thank you for your comment. We modified "at similar heights" to "at the almost same height" line 1 on page 6 in the revised manuscript.

6. The reflectivity was calculated from the DSD measurements at 1 min resolution -> The reflectivity was calculated from DSD at 1 min resolution

Author's Response:

Thank you for your kind comment. We modified the sentence as reviewer's comment from line 24 to 25 on page 6 in the revised manuscript.

7. 3.1 Equidistance line method, concerning Fig. 6, it shows average reflectivity difference between PSN and BSL. Which radar shows higher reflectivity? What is the reason on the dominant peak of the average value? It should be explained.

Author's Response:

Thank you for your comment. The negative bias means that the PSN reflectivity was underestimated comparing with BSL reflectivity. For better understanding, we added the following sentence from line 29 to 30 on page 6 in the revised manuscript. "The reflectivity observed by BSL or PARSIVEL subtracted from that observed by PSN was taken as a $Z_H$ bias." And we also added the following description from line 26 to 27 on page 8 in the revised manuscript. "It means that the reflectivity observed by PSN was underestimated comparing with BSL." The dominant peak of the average value occurred from 1500 LT would be caused by the decreased sample number for the comparison of both reflectivities. Therefore, we added the following sentence from line 28 to 31 on page 8 in the revised manuscript. "The number of samples was generally above 60, but it was smaller than 60 after 1450 LST. The dominant peak of the averaged reflectivity difference occurred from 1500 LST would be caused by the decreased sample number for the comparison of reflectivity observed from both radars."

8. 3.1 Equidistance line method, comparing the average reflectivity differences between PSN and BSL in Fig. 6 and 7, most of the average values in Fig. 7 is less than 0 dB. However, most of the average values in Fig. 6 is higher than 0 dB. It should be explained in detail.

Author's Response:

Thank you for your comment. Figure 6 shows the time series of the average reflectivity

difference between PSN and BSL on 25 August in 2014 and Figure 7 is same as Fig. 6 but for on 8 September 2012. The rainfall system was different from each other. We expect that the difference came from two reasons, one is different precipitation system and the other is radar hardware calibration accuracy. Therefore, we added the following sentence from line 20 to 23 on page 11 in the revised manuscript. "It would be caused by the difference of total rainfall amount between two rainfall systems. The maximum total rainfall amount for both cases were around 250 mm for 25 August and 150 mm for 8 September 2012. Another reason of the fluctuation would be the difference of radar hardware calibration error for PSN between two events."

9. 3.2 Overlapping area method, concerning Fig. 10, it shows average reflectivity difference between PSN and BSL. Which radar shows higher reflectivity? What is the reason on the dominant peak of the average value around 03 LST? It should be explained.

Author's Response:

Thank you for your comment. The negative averaged reflectivity means that BSL reflectivity is higher than that of PSN because the difference is equal to the values which are BSL reflectivity subtracted from PSN reflectivity. The possible reason on the dominant peak of the average value around 03 LST is that the sudden change of microphysical characteristics of rainfall pass through the overlapping area. Therefore, we added the following sentence from line 20 to 22 on page 9 in the revised manuscript. "The fluctuation also would be caused by the sudden change of microphysical characteristics of rainfall pass through the overlapping area for both radars. It would reduce the accuracy of $Z_H$ of BSL corrected by self-consistency"

10. 3.3 Disdrometer method, what are the daily rainfall amounts from the gauges and Parsivel on 8 Sept. 2012 ?

Author's Response:

**AMTD**

Thank you for your comment. We added the daily rainfall amounts from both instruments from line 9 to 10 on page 10 in the revised manuscript as follows; "daily total rainfall amounts for PARSIVEL and the gauge were 54.4 and 55.0 mm, respectively."

11. 3.3 Disdrometer method, explain the reason on the unstable behavior of reflectivity from Parsivel in Fig. 13. Is it reasonable to use Parsivel data as a reference in this study? Why is it sufficiently reliable to us as a reference in split of the unstable behavior.

Author's Response:

Thank you for your comments. We expect that the unstable behavior of reflectivity from Parsivel from 1400 LST to 1500 LST in Fig. 13 of original manuscript was caused by the sudden change of rainfall at that time period. We also should consider the threshold of reflectivity value observed from both PSN and PSRSIVEL to get more reliable $Z_H$ bias. The bias would be obtained more accurately when the reflectivity values observed from both instruments were higher than 15 dBZ in this event. Therefore, we added the following sentence from line 17 to 21 on page 10 in the revised manuscript. "The sudden change of rainfall would cause the unstable reflectivity difference from 1340 LST to 1500 LST. The threshold of reflectivity value observed from both PSN and PSRSIVEL should be considered for the comparison to get more reliable $Z_H$ bias. The bias would be obtained more accurately when the reflectivity values observed from both instruments were higher than 15 dBZ in this event." Regarding with the possibility of PARSIVEL usage for getting $Z_H$ bias, there are two main uncertainties come from the wind effect and the difference height between both instruments. To know the accuracy of disdrometer, 10 mins rainfall amount measured from disdrometer and gage were compared. We found out that rainfall amount obtained from disdrometer was correlated to that of gage as mentioned in the manuscript. We did quality control algorithm using the equation suggested by Jaffrain and Berne (2011) to reduce the wind effect. $|v(D)_{meas}\text{-}v(D)_{Beard}|{\leq}0.6v(D)_{Beard}$

Where, $v(D)_{meas}$ is the velocity measured by PARSIVEL, $v(D)_{Beard}$ is the velocity for a

drop diameter D according to Beard's model. We checked the maximum wind speed for all cases were less than 8 ms$^{-1}$. Friedrich and Higgings (2013) found out that once the wind speed exceeded a critical value, approximately 15-20 ms$^{-1}$ based on the observations during Hurricane Ike and VORTEX2, the PARSIVEL continuously observed unrealistically large slow-falling drops as seen during Hurricane Ike. Therefore, we think that the disdrometer data can be used for the analysis. Anyway, we added the limitations from line 28 to 30 on page 11 in the revised manuscript as follows; "It is worth to noting that the result would be changed when the drop size distributions was fluctuated with height especially at the layer between radar beam and ground in case of disdrometer method."

12. 4 Conclusion, the authors should include comparison of three methods using some statistical parameter.

Author's Response:

Thank you for your comment. We added the following sentences from line 9 to 26 on page 12 in the revised manuscript "The rainfall estimation using $Z = 200R^{1.6}$ and $Z=300R^{1.4}$ and gauge rainfall were examined for 25 August 2014 and 8 September 2012 to investigate the accuracy of each method. The RMSE, NE, and CC of rainfall pairs for $Z = 200R^{1.6}$ ($Z=300R^{1.4}$) on 25 August 2014 in case of using equidistance method were improved from 65.7 (66.1) to 32.6 (27.0) mm, from 0.79 (0.81) to 0.36 (0.31), and from 0.88 (0.87) to 0.89 (0.88), respectively. On 8 September 2012, the RMSE, NE, and CC for $Z = 200R^{1.6}$ ($Z=300R^{1.4}$) changed from 30.0 (28.5) to 22.5 (20.0) mm, from 0.58 (0.56) to 0.41 (0.36), and from 0.81 (0.8) to 0.78 (0.76), respectively. The RMSE and NE of rainfall pairs for $Z = 200R^{1.6}$ ($Z=300R^{1.4}$) on 25 August 2014 in case of using overlapping method were improved from 65.7 (66.1) to 29.7 (25.8) mm and from 0.79 (0.81) to 0.31 (0.28), respectively. On 8 September 2012, RMSE and NE for $Z = 200R^{1.6}$ ($Z=300R^{1.4}$) were improved from 30.0 (28.5) to 21.8 (19.1) mm and from 0.58 (0.56) to 0.40 (0.34), respectively, by the use of bias correction, while CC for $Z=200R^{1.6}$ was unchanged at 0.81 and that of $Z=300R^{1.4}$ were
changed 0.8 to 0.79. The RMSE and NE of rainfall pairs for $Z = 200R^{1.6}$ ($Z=300R^{1.4}$) on 25 August 2014 in case of using disdrometer method were improved from 65.7 (66.1) mm to 42.0 (61.4) mm and from 0.79 (0.81) to 0.40 (0.53), respectively. On 8 September 2012, RMSE and NE for $Z = 200R^{1.6}$ ($Z=300R^{1.4}$) decreased from 30.1 (28.6) to 24.6 (23.9) mm, and from 0.58 (0.56) to 0.46 (0.44), respectively, while CC for $Z = 200R^{1.6}$ ($Z=300R^{1.4}$) decreased from 0.81 (0.8) to 0.65 (0.59)."

B. Reviewer's recommendation

1. This paper shows radar reflectivity bias correction using three methods to get more accurate rainfall from single polarization radar. The approaches and results are considered as reasonable. Therefore, this paper is recommended for its publication with corrections as suggested in the major comments.

Author's Response:

Thank you for your kind comments. We revised the manuscript according to the reviewer's all comments. We also described the self-consistency method for ZH bias correction using BSL radar, added the hourly reflectivity distribution for both rain events used for the analysis, and added the results from rainfall estimation using $Z=300R^{1.4}$ to follow the reviewer 1's comments. We also modified some mistakes like misspelling.

\*\*\* Thank you very much again for your deep review and it will be of much help for better our manuscript quality.\*\*\*

Please also note the supplement to this comment:
http://www.atmos-meas-tech-discuss.net/amt-2015-392/amt-2015-392-AC2-supplement.pdf

---

## Referee Comment (RC3) · Anonymous Referee #3 · 28 Mar 2016

**Approaches to radar reflectivity bias correction to improve rainfall estimation in Korea.**

March 28, 2016

**Summary**

This study addressed a major problem associated with weather radar data. Three methods are proposed for radar reflectivity bias correction. Three methods are proposed for reducing errors in single polarimetric radar (SPOL) reflectivity using either polarimetric radar (DPOL) and or DSD measurements. The DSD information is derived by a disdrometer located 9 km away from PSN. First method tries to correct the data using the reflectivities on the equidistance line between the two radar devices. The second approach using the overlapping area of the two radars for correction purposes. The third approach proposes correcting radar data by the reflectivity derived from the DSD observations of a disdrometer. Two events in 2014 and 2012 are selected to investigate the techniques. The validation is carried out by comparing the observations of 121 gauges and radar rainfall estimates. All the three methods improved the accuracy of estimated rainfall, except for a period when DSDs were not observed. This was explained by the fact that the disdrometer was not able to observe the entire rain event.

**General comments**

Although the content of the paper is relevant to the meteorological community, there are some concerns. The philosophical justification for each approach is not clear. The DPOL provide more accurate information than SPOL, but still suffers from errors. However, why should a source containing errors (DPOL) be used for correcting another source (SPOL). Even if the approach is justifiable, why should one correct a source where a better source of data exists. I mean, one can use directly DPOL data instead of taking the effort for correcting SPOL? Another point is that the DPOL device is an S-Band radar system. How about the other device? Because of the radar radius, I assume that the SPOL device should be an C-Band system, (Figure 4, left)? We know that S-Band has unique problems. How do you give explanation for using two different devices for correction? If my guess is not valid, the authors must provide a better explanation of the two devices. When it comes to using disdrometer data and the equidistance method, it is not clearly explained how it is extended to the rest of the area. A better description must be provided.

In general, I suggest to compare the methods with a common correction method for a better conclusion and investigation of the techniques. Furthermore, the validation must be more clear. What is the temporal resolution of the data being evaluated? How do you explain comparing radar data with point data. How do you justify using a constant parameter set for the Z-R relationship?

**Specific comments**

**P1, L22 to L24**: You talk about combining all the three methods. You must address a way how to combine them, and if you think it is better, you should add it to the paper as the fourth method.

**P2, L6 to L8:** How do you evaluate each method in these regards? Your reference data is uncertain!

**P3:** why don't you separate the two sections of "Data" and "Methodology". Each method should be described separately in a subsection. A better description of data must be provided.

**P3, L9 to L13:** You start the paragraph with "Data". What are you referring to? The entire paragraph is a bit unclear.

**P3, L14:** What is PSN? What is BSL? Explain the location of each radar device and the abbreviations.

**P3, L30:** You must give reasons for taking the equidistance line for correction. How do you use the correction for the rest of the study area? How reliable is the approach?

**P5, L8:** "reflectivity and $Z_{DR}$" **-** Either both symbols or both the entire word.

 **P5, L9:** What is the "systematic bias"?

**P6, L16 to L19:** The events must be described in the "Data" section. You should explain a bit the types of the two events.

**P6, L21:** What are the "reflectivity biases"?

**P7, L9:** What is the "precipitation system"?

**P8, S3.3:** As already asked, it is not clear how you use this information for the rest of the study area. For example, for the points far away from the disdrometer.

**P9, L1:** Which one is "Fig. 16a)"?

**P14:** What are the circle? The legend must be provided including the scaling. What are BSL and PSN and AWAS? Those must be also described in the text.

**P15:** A similar question to the correction approach using disdrometer data. How do you use the information for the rest of the study area?

**P17:** What are the circles? The gray areas?

**P18:** You are averaging over 3 km × 3°. How do you then take the spatial bias into consideration?

**P21:** Following the y-intercept, the given equations are not going through the origin. How is it possible? Would it not result in a systematic error?

**P22, and P23:** A complete description of the figure must be provided.

**Technical corrections**

There are some parts with poor English. The ones I found:

**P1:** The last sentence in the abstract.

**P2: L4-L5**

---

## Author Comment (AC3) · 31 Mar 2016

Response to review At first, thank you very much for referee's effort in reviewing our paper even your busy time. We revised the manuscript titled "Approached to radar reflectivity bias correction to improve rainfall estimation in Korea" that was submitted to Atmospheric Measurement Techniques. The manuscript has been revised as suggested by reviewer and we also corrected some mistakes. We would appreciate any feedback on the revisions.

Response to review by Anonymous referee 3

General comments

1. Although the content of the paper is relevant to the meteorological community, there

are some concerns. The philosophical justification for each approach is not clear. The DPOL provide more accurate information than SPOL, but still suffers from errors. However, why should a source containing errors (DPOL) be used for correcting another source (SPOL). Even if the approach is justifiable, why should one correct a source where a better source of data exists. I mean, one can use directly DPOL data instead of taking the effort for correcting SPOL?

Author's Response: Thank you for your comment. As reviewer's comment, the best way is to use DPOL directly however, there is some cases SPOL and DPOL are operated at the same time. For example, KMA (Korea Meteorological Administration) is replacing 10 SPOLs to DPOL year by year. The time period when SPOL and DPOL are operated at the same time should be existed for a few years. There is no way to correct reflectivity biases of SPOL, however, we could correct the reflectivity biases of DPOL using self-consistency method. As mentioned in the manuscript, the reflectivity of SPOL was much underestimated. If we use the reflectivity to calculate radar rainfall, its accuracy would not be reliable. When we would like to calculate more accurate climatological radar (SPOL) rainfall, we have to correct the reflectivity errors using possible methods at first. And we would also use this technique in real time for estimating radar rainfall in case both SPOL and DPOL are operated at the same time.

2. Another point is that the DPOL device is an S-Band radar system. How about the other device? Because of the radar radius, I assume that the SPOL device should be an CBand system, (Figure 4, left)? We know that S-Band has unique problems. How do you give explanation for using two different devices for correction? If my guess is not valid, the authors must provide a better explanation of the two devices.

Author's Response: Thank you for your comment and we are sorry to make confused. Both radars are S-band radars. We added the frequency of PSN radar in the revised manuscript 15 line on page 3. "the frequency is 2.712 GHz"

3. When it comes to using disdrometer data and the equidistance method, it is not

clearly explained how it is extended to the rest of the area. A better description must be provided.

Author's Response: Thank you for your comment and we are sorry for the confusion. Once the reflectivity bias is calculated, the bias will be applied to all the pixels of SPOL coverage. Because all three approaches used in this study including disdrometer and the equidistance method is to obtain the bias of SPOL reflectivity. We added the following sentence from 23 to 25 line on page 7 in the revised manuscript for better understanding. "The reflectivity observed by BSL or PARSIVEL subtracted from that observed by PSN was taken as a $Z_H$ bias and it will be applied to all pixels of PSN coverage."

4. In general, I suggest to compare the methods with a common correction method for a better conclusion and investigation of the techniques.

Author's Response: Thank you for your comments. Actually, we could not find the common correction method of SPOL reflectivity in the previous literatures. That is why we tried to propose three approaches to correct SPOL reflectivity. We think that three approaches used in this study would be possible way to correct SPOL reflectivity.

5. Furthermore, the validation must be more clear. What is the temporal resolution of the data being evaluated?

Author's Response: Thank you for your comments. We added the following sentences line 29 on page 7 and line 1 on page 8 and line 8 on page 8 in the revised manuscript for better understanding. Line 29 on page 7 and 1 on page 8 : "The rain gages were 0.5 mm tipping-bucket type. Time resolution of gages is 1 min and data quality control was done by KMA." Line 8 on page 8 : "total accumulated rainfall amounts for analyzed time period"

6. How do you explain comparing radar data with point data.

Author's Response: Thank you for your comments. As reviewer's comment, there would be some errors on comparing areal data (radar) with point data (gage). We tried to reduce the error by taking small area data (1 km $\times$ 1°) from radar centered on the corresponding rain gauge. Most studies on validation of radar rainfall estimation, gage rainfall is used as a reference. Please understand this.

7. How do you justify using a constant parameter set for the Z-R relationship?

Author's Response: Thank you for your comments. As reviewer's comment, there are many uncertainties of radar rainfall estimation. The variability of Z-R relation is one of main source of these uncertainties. The Z-R relations are different from storm to storm, precipitation types, climatology, and so on. In this study, we would like to show that if the reflectivity is corrected adequately, rainfall estimation would be improved. Anyway we added the results of validations obtained by another Z-R relation, $Z=300R^{1.4}$ which is widely used for NEXRAD in the revised manuscript for better understanding.

Specific comments

P1, L22 to L24: You talk about combining all the three methods. You must address a way how to combine them, and if you think it is better, you should add it to the paper as the fourth method.

Author's Response: Thank you for your comment. As reviewer's comment, the sentences are not connected to our result directly. We removed the lines in the revised manuscript as reviewer's comment.

P2, L6 to L8: How do you evaluate each method in these regards? Your reference data is uncertain!

Author's Response: Thank you for your comment. We put the sentences just to explain the difficulties on radar rainfall estimation in general not to evaluate all things in this study.

P3: why don't you separate the two sections of "Data" and "Methodology". Each method should be described separately in a subsection. A better description of data must be provided.

Author's Response: Thank you for your comment. We divided "Data and Methodology" into two separated section and we described each method in a subsection in the revised manuscript as reviewer's comment. For better description of data, we added Figures 2, 3 and the following sentences to describe the rainfall cases from line 1 to 23 on page 4 in the revised manuscript." Figure 2 shows the time series of $Z_H$ observed from BSL radar on 8 September in 2012 and 25 August in 2014. The precipitation within radar coverage on 8 September in 2012 was occurred by low pressure with the front located at northern part of Korea. The core of the precipitation systems was elongated from south to north and moved to eastward. The maximum reflectivity of the core was more than 45 dBZ and caused rainfall at the western part of radar center at 0300 LST (Fig. 2(a)), became more organized shape at the eastern part of radar center at 0400 LST (Fig. 2(c)), and moved to eastward and located out of land at 0500 LST (Fig. 2(e)) on 8 September in 2012. The precipitation system on 25 August in 2014 was caused by the low pressure located at southern part of Korea. The two strong rainfall within the radar coverage were located at south-western part of radar center with distance between 120 km and 150 km and southern part of radar center with distance between 30 km and 90 km, respectively at 1200 LST on 25 August in 2014 (Fig. 2(b)). The two convective cells moved to eastward, their strength were intensified and the area of rainfall was wider at 1300 LST (Fig. 2(d)). The two systems moved to eastward continuously, were merged together at 1400 LST (Fig. 2(f)). Figure 3 shows the time series of hourly rainfall and daily accumulation measured by a gage which recorded highest daily rainfall within radar coverage on 8 September in 2012 and 25 August in 2014. The highest daily accumulated rainfall was recorded from North Changwon (ID 255) and Geumjeong (ID 939) on each day, respectively. The daily accumulation of ID 255 was 150 mm, the maximum hourly rainfall was around 40 mm, and the duration of the rainfall was 7 hours (Fig. 3(a)). The daily accumulation of ID 939 was around 270 mm, the maximum hourly rainfall was more than 100 mm h-1. The rainfall amount for 3 hours (1000 LST, 1400 LST, and 1500 LST) were mainly contributed to the total rainfall
accumulation on 25 August in 2014 (Fig. 3(b))."

P3, L9 to L13: You start the paragraph with "Data". What are you referring to? The entire paragraph is a bit unclear.

Author's Response: Thank you for your kind comment. We modified "Data" to "Data observed from PARSIVEL".

P3, L14: What is PSN? What is BSL? Explain the location of each radar device and the abbreviations.

Author's Response: Thank you for your kind comment and we are sorry for confusion. We added the abbreviations line 9 and 10 on page 3 in the revised manuscript as follows; "PSN (Pusan radar) is located at coastal line and BSL (Bisalsan radar) is located 76.9 km away from PSN (Fig. 1),"

P3, L30: You must give reasons for taking the equidistance line for correction. How do you use the correction for the rest of the study area? How reliable is the approach?

Author's Response: Thank you for your comment. The comparison of both reflectivity observed from PSN and BSL which point same target was done in order to get more reliable results. That is why we tried to find out the equidistance line for both radars. And then the difference of reflectivity at the equidistance line would be considered as a systematic bias. Once the bias is obtained, the bias will be applied to all pixels of PSN coverage. As mentioned in the manuscript, the equidistance method would be used if the sample number is enough.

P5, L8: "reflectivity and $Z_{DR}$" - Either both symbols or both the entire word.

Author's Response: Thank you for your comment. We modified "reflectivity and $Z_{DR}$" to "$Z_H$ and $Z_{DR}$" line 26 on page 4 in the revised manuscript.

P5, L9: What is the "systematic bias"

Author's Response: Thank you for your comment. The errors are composed of systematic bias and random error. We could correct systematic bias but not random error. We focused on the systematic bias. Anyway, for better understanding we modified "systematic bias" to "bias".

P6, L16 to L19: The events must be described in the "Data" section. You should explain a bit the types of the two events.

Author's Response: Thank you for your comment. We moved the sentences in the "Data" section and added explanation of two events in the revised manuscript as mentioned in the author's response to the previous comment.

P6, L21: What are the "reflectivity biases"?

Author's Response: Thank you for your comments. I assume that reviewer means "reflectivity bias" at line 13 on page 6 in the original manuscript. The reflectivity bias means the difference reflectivity between PSN and BSL (PARSIVEL). We modified "reflectivity bias" to "the difference reflectivity between PSN and BSL (PARSIVEL)" line 28 and 29 on page 3 in the revised manuscript for better understanding.

P7, L9: What is the "precipitation system"?

Author's Response: Thank you for your comment. We modified the sentence to the following sentence line 16 to 18 on page 9 in the revised manuscript. "This result suggests that the rainfall observed from both BSL and PSN radar was not located enough over the equidistance line to get a reliable comparison until 0310 LST."

P8, S3.3: As already asked, it is not clear how you use this information for the rest of the study area. For example, for the points far away from the disdrometer.

Author's Response: Thank you for your comment. We added the following sentence from 23 to 25 line on page 7 in the revised manuscript for better understanding. "The reflectivity observed by BSL or PARSIVEL subtracted from that observed by PSN was taken as a $Z_H$ bias and it will be applied to all pixels of PSN coverage."

[Figure]

P9, L1: Which one is "Fig. 16a)"?

Author's Response: Thank you for your comment and we are sorry for confusion. We added (a) and (b) in Figure 18 (as Figure 16 in original) of revised manuscript.

P14: What are the circle? The legend must be provided including the scaling. What are BSL and PSN and AWAS? Those must be also described in the text.

Author's Response: Thank you for your comment. We modified figure and captions in Figure 1.

P15: A similar question to the correction approach using disdrometer data. How do you use the information for the rest of the study area?

Author's Response: Thank you for your comment. We added the following sentence from 23 to 25 line on page 7 in the revised manuscript for better understanding. "The reflectivity observed by BSL or PARSIVEL subtracted from that observed by PSN was taken as a $Z_H$ bias and it will be applied to all pixels of PSN coverage."

P17: What are the circles? The gray areas?

Author's Response: Thank you for your comment. We added the following sentence in the caption of the Figure 4 in the revised manuscript. "The red (blue) dotted circle shows the maximum range of BSL (PSN) and gray shaded area show 200 km by 200 km extracted from each radar coverage in the left panel."

P18: You are averaging over 3 km $\times$ 3°. How do you then take the spatial bias into consideration?

Author's Response: Thank you for your comment. As mentioned previous reviewer's comments, we would like to get systematic bias not random noise using equidistance line, overlapping area and disdrometer methods. We assumed the calculated difference between PSN and BSL (PARSIVEL) as a systematic bias. Once the bias is calculated, the bias will be then applied to all pixels of radar coverage.

P21: Following the y-intercept, the given equations are not going through the origin. How is it possible? Would it not result in a systematic error?

Author's Response: Thank you for your comment. The fitting line shows the relation between radar rainfall and aws rainfall. If both rainfall are completely same, the given equations are going through the origin. However, the rainfall from radar is not same as that from gage. We agreed with reviewer's comment. It would be caused by a systematic error.

P22, and P23: A complete description of the figure must be provided.

Author's Response: Thank you for your comment. We added complete description of the figure.

Technical corrections There are some parts with poor English. The ones I found: P1: The last sentence in the abstract.

Author's Response: Thank you for your comment. As reviewer's comment, we have already removed the sentence.

P2: L4-L5

Author's Response: Thank you for your comment. We modified the sentence to the following sentence in the revised manuscript. "There is no unique R(Z), since DSDs can be varied storm to storm and even within a single storm (Battan 1973; You et al., 2010)."

Additional revision according to other reviewers' comment 1. We added some sentences to describe the self consistency method from line 12 on page to line 4 on page in the revised manuscript.

2. We added some sentences in Sect. 5 from line 9 to 26 on page 12 in the revised manuscript.

3. And we also modified or added some sentences in the revised manuscript for better understanding.

*** Thank you very much again for your deep review and it will be of much help for better our manuscript quality.***

Please also note the supplement to this comment:
http://www.atmos-meas-tech-discuss.net/amt-2015-392/amt-2015-392-AC3-supplement.pdf

**Supplement:**

[revised manuscript text omitted]

The quality controlled $Z_H$, $Z_{DR}$, $K_{DP}$ measured from BSL were used to calibrate $Z_{DR}$ and $Z_H$ of

BSL. The $Z_H$ measured from PSN were then corrected by using calibrated $Z_H$ of BSL using self-consistency method and $Z_H$ measured by PARSIVEL. The gage rainfall data were used to assess the performance of three $Z_H$ bias correction methods for PSN which is SPOL.

The accuracy of rainfall estimation using corrected reflectivity was evaluated to measure the effectiveness of each method for calculating the difference reflectivity between PSN and BSL

(PARSIVEL). Two rainfall events were used, occurring on 25 August 2014 and 8 September

2012 (Table 1). The August and September events were caused by low pressure systems over southern and northern Korea, respectively.

Figure 2 shows the time series of $Z_H$ observed from BSL radar on 8 September in 2012 and 25 August in 2014. The precipitation within radar coverage on 8 September in 2012 was occurred by low pressure with the front located at northern part of Korea. The core of the precipitation systems was elongated from south to north and moved to eastward. The maximum reflectivity of the core was more than 45 dBZ and caused rainfall at the western part of radar center at 0300 LST (Fig. 2(a)), became more organized shape at the eastern part of radar center at 0400 LST (Fig. 2(c)), and moved to eastward and located out of land at 0500 LST (Fig. 2(e)) on 8 September in 2012. The precipitation system on 25 August in 2014 was caused by the low pressure located at southern part of Korea. The two strong rainfall within the radar coverage were located at south-western part of radar center with distance between 120 km and 150 km and southern part of radar center with distance between 30 km and 90 km, respectively at 1200 LST on 25 August in 2014 (Fig. 2(b)). The two convective cells moved to eastward, their strength were intensified and the area of rainfall was wider at 1300 LST (Fig. 2(d)). The two systems moved to eastward continuously, were merged together at 1400 LST (Fig. 2(f)).

Figure 3 shows the time series of hourly rainfall and daily accumulation measured by a gage which recorded highest daily rainfall within radar coverage on 8 September in 2012 and 25 August in 2014. The highest daily accumulated rainfall was recorded from North Changwon (ID 255) and Geumjeong (ID 939) on each day, respectively. The daily accumulation of ID 255 was 150 mm, the maximum hourly rainfall was around 40 mm, and the duration of the rainfall was 7 hours (Fig. 3(a)). The daily accumulation of ID 939 was around 270 mm, the maximum hourly rainfall was more than 100 mm h$^{-1}$. The rainfall amount for 3 hours (1000 LST, 1400 LST, and 1500 LST) were mainly contributed to the total rainfall accumulation on 25 August in 2014 (Fig. 3(b)).

**3 Methodology**

**3.1 Z and $Z_{DR}$ bias correction for BSL**

Before calculating reflectivity bias for PSN using BSL, $Z_H$ and $Z_{DR}$ must be corrected for bias. $Z_{DR}$ bias correction is important for the absolute calibration of the radar using a self-consistency method. Gorgucci et al. (1999) proposed using a vertical pointing scan of light rain, to take advantage of the nearly spherical shape of the raindrops as seen from below. Ryzhkov et al (2005) used the elevation angle dependency of $Z_{DR}$ as an alternative technique and concluded that the high variability of $Z_{DR}$ in rainfall prohibited the method from achieving the required absolute calibration accuracy of 0.2 dB. They instead proposed a method that utilizes the structural characteristics of the melting layer in stratiform clouds and the dry aggregated snow present above the melting layer. $Z_{DR}$ measurements from dry aggregated snow above the melting layer resulted in a mean S-band value of 0.2 dB and an accuracy of 0.1–0.2 dB. Trabal et al. (2009) evaluated two methods using the intrinsic properties of dry aggregated snow present above the melting layer and light rain measurements close to the ground, and found that a $Z_{DR}$ calibration accuracy of 0.2 dB or better was achieved using either method.

Vertical pointing data were not available in the present case, and the scan strategy, with six elevation angles, was unable to detect the melting layer. Therefore, in this study, light rain measurements close to the ground were used to calibrate $Z_{DR}$. Light rain was defined using a threshold of $20\ dBZ \le Z \le 28\ dBZ$, as proposed by Marks et al. (2011). The assumption of $Z_{DR}$ is close to zero in case of the small rain drop like drizzle was chosen for this study. The $Z_{DR}$ observed from BSL having with reflectivity in the range of 20 dBZ to 28 dBZ for given time period were averaged. Then the averaged $Z_{DR}$ was taken as a $Z_{DR}$ bias.

The $Z_H$ bias was calculated by self-consistency method using a 9-gate moving average of bias corrected $Z_{DR}$ in the range of 0.2 dB to 3.0 dB to improve the accuracy. This method depends on the notion that $Z_H$, $Z_{DR}$, and $K_{DP}$ are independent in rain, and that $Z_H$ can be estimated from $Z_{DR}$ and $K_{DP}$. The difference between the computed and observed values of $Z_H$ is referred to as the Z bias. Following the method of Ryzhkov et al. (2005), the entire spatial and temporal domain was divided into 1 dB intervals of $Z_H$ between Zmin (30 dBZ) and Zmax (50 dBZ), and the $K_{DP}(Z_H)$ and $Z_{DR}(Z_H)$ within each interval were calculated. The $Z_H$ bias is then determined by matching the integrals:

$$I_1 = \sum_{Z_{min}}^{Z_{max}} K_{DP}(Z)n(Z)\Delta Z , \tag{1}$$

$$I_2 = \sum_{Z_{min}}^{Z_{max}} 10^{0.1Z_m} f(Z_{DR})n(Z)\Delta Z , \tag{2}$$

The function of f($Z_{DR}$) in Eq. (2) can be well approximated by a fourth-order polynomial fit for certain range of $Z_{DR}$ (Gourley et al., 2009) like Eq. (3).

$$f(Z_{DR}) = 10^{-5}(a_0 + a_1 Z_{DR} + a_2 Z_{DR}^2 + a_3 Z_{DR}^3) , \tag{3}$$

The estimated $Z_H$ bias is determined from Vivekanandan et al. (2003) by

$Z_H bias(dB) = 10 \log(\frac{I_2}{I_1})$,                   (4)

If the radar is well calibrated, $Z_H$ bias should be equal to 0. The coefficients of $f(Z_{DR})$ were calculated by T-matrix scattering method using long period DSD data and are 4.26, -4.67, 2.67, and -0.54, respectively.

**3.2    Equidistance line method**

To calculate the reflectivity bias of PSN, which is single polarization radar, three approaches were used: the equidistance line method, the overlapping area method, and the disdrometer method. The first approach is to compare the reflectivities along the line that is equidistant between the two radars. To determine this line for the two radars, the effective radius was set to 100 km and the distance between the two radars and the azimuthal angle pointing from BSL

to PSN were calculated using their latitude and longitude values. The start and end azimuthal angles for comparison of reflectivity were then calculated as follows:

$AZ_{st} = \beta - a\cos(0.5 \times dr / rc)$                  (5)

$AZ_{end} = \beta - a\cos(0.5 \times dr / rc) + 2 \times a\cos(0.5 \times dr / rc)$,       (6)

where $AZ_{st}$ and $AZ_{end}$ are the start and end azimuthal angles for the comparison, respectively; $\beta$

is an azimuthal angle which is the angle between north and the bearing from BSL points to

PSNand $rc$ and $dr$ are the effective radius and distance from BSL to PSN, respectively. The distance between the two radars is 76.9 km, and the start and end azimuthal angles of BSL (PSN)

are 79 (35) and 213 (261) degrees, respectively (Fig. 4).

To compare the reflectivity observed of targets at the almost same height from both radars, the beam height was calculated assuming a standard atmospheric beam propagation (Rinehart,

2010), as follows:

$H = \sqrt{r^2 + (R'+H_0)^2 + 2r(R'+H_0)\sin\phi} - R'$,          (7)

where $r$ is the slant range from the radar, $\Phi$ is the elevation angle of the radar beam, $H_0$ is the height of the radar antenna above sea level, and R' = (4/3)R, where R is the Earth's radius (6,371 km). The radar antenna heights of PSN and BSL are 547 and 1085 m, respectively.

Figure 5 shows the beam height of PSN with blue solid line and BSL at the equidistance line (blue dashed line as shown in Fig. 4). EL1 to EL6 show the elevation angles from smallest to largest. The smallest difference in beam height between the two radars is 149 m, which was obtained using the fourth elevation angle of PSN and the third elevation angle of BSL. Therefore, the reflectivity bias of PSN was calculated by averaging the difference of reflectivity along with the equidistance line observed from fourth elevation angle of PSN and third one of BSL.

**3.3 Overlapping area method**

In the second approach, the overlapping area for the two radars was calculated by matching the coordinates. The polar coordinate of two radars was converted to a Cartesian coordinate with a spatial resolution of 1 km. The overlapping area was then determined by considering the distances between the two radars in the east–west and north–south directions. Figure 6 shows a schematic diagram of the overlapping area for the two radars. The distance between two radars in east-west and north-south direction are 42 km and 64 km, respectively. The reflectivity observed from both radars at the pixels designated at the overlapping area as shown by blue rectangle in right panel of Fig. 6 were compared to calculate the $Z_H$ bias of PSN. The extracted domain of PSN and BSL for the comparison is $158 \times 136$ km.

**3.4 Disdrometer method**

The third and final approach is to use DSD observations from the PARSIVEL disdrometer. The reflectivity was calculated from the DSD at 1 min resolution, and averaged over 10 mins to match the radar time resolution. Figure 7 shows a schematic of the procedure used to match the radar and PARSIVEL data. The PARSIVEL disdrometer is located ~9 km from the radar, at an azimuthal angle of 87 degrees. The radar reflectivity was averaged over a domain of 13 gates $\times$ 3 degrees in azimuth, centered at the PARSIVEL location. The reflectivity observed by BSL or PARSIVEL subtracted from that observed by PSN was taken as a $Z_H$ bias and it will be applied to all pixels of PSN coverage.

**3.5 Validation**

The normalized error (NE), root-mean-square error (RMSE), and correlation coefficient (CC) between rainfall estimates and measurements from 121 gauges were calculated to measure the performance of each bias correction method. The rain gages were 0.5 mm tipping-bucket type.

Time resolution of gages is 1 min and data quality control was done by KMA. These quantities are defined as follows:

$$NE = \frac{\frac{1}{N}\sum_{i=1}^{N}\left|R_{R,i} - R_{G,i}\right|}{\overline{R_G}} \tag{8}$$

$$RMSE = \left[\frac{1}{N}\sum_{i=1}^{N}(R_{R,i} - R_{G,i})^2\right]^{1/2} \tag{9}$$

$$CC = \frac{\sum_{i=1}^{N}(R_{R,i} - \overline{R_R})(R_{G,i} - \overline{R_G})}{\left[\sum_{i=1}^{N}(R_{R,i} - \overline{R_R})^2\right]^{1/2}\left[\sum_{i=1}^{N}(R_{G,i} - \overline{R_G})^2\right]^{1/2}}, \tag{10}$$

where N is the number of radar rainfall ($R_R$) and gauge rainfall ($R_G$) pairs, and $\overline{R_R}$ and $\overline{R_G}$ are the average hourly rain rates from radar and gauges, respectively. These quantities were calculated using total accumulated rainfall amounts for analyzed time period from radar and gauge measurements at each point. The radar rainfall value at each point was obtained by averaging rainfall over a small area (1 km × 1°) centered on the corresponding rain gauge. The radar rainfall was calculated using the relation $Z = 200\ R^{1.6}$ and $Z = 300\ R^{1.4}$.

**4 Results**

**4.1 Equidistance line method**

Before estimating radar rainfall rates, reflectivity biases were calculated using each of the three methods. Figure 8 shows time series of the average reflectivity difference between PSN and BSL at the equidistance line and the number of samples used in each calculation, on 25 August 2014. The average difference over the entire time period was −7.85 dB, and the largest difference was −12.46 dB. It means that the reflectivity observed by PSN was underestimated comparing with BSL. The number of samples used for each calculation was determined using a beam height difference threshold of 0.1 km. The number of samples was generally above 60, but it was smaller than 60 after 1450 LST. The dominant peak of the averaged reflectivity difference occurred from 1500 LST would be caused by the decreased sample number for the comparison of reflectivity observed from both radars. Figure 9 shows the same information for 8 September 2012. The average reflectivity difference over the entire time period was − 2.56 dB, and the largest difference was −6.77 dB. The number of samples was less than 50 until 0310 LST, after which it increased to more than 50. This result suggests that the rainfall observed from both BSL and PSN radar was not located enough over the equidistance line to get a reliable comparison until 0310 LST.

Figure 10 shows the scatter plot of total accumulated radar rainfall amount for analyzed time period, calculated using $Z = 200R^{1.6}$ and $Z=300R^{1.4}$ and gauge rainfall, for 25 August 2014 and 8 September 2012. The RMSE, NE, and CC of rainfall pairs for $Z = 200R^{1.6}$ ($Z=300R^{1.4}$) on 25 August 2014 were improved from 65.7 (66.1) to 32.6 (27.0) mm, from 0.79 (0.81) to 0.36 (0.31), and from 0.88 (0.87) to 0.89 (0.88), respectively. On 8 September 2012, the RMSE, NE, and CC for $Z = 200R^{1.6}$ ($Z=300R^{1.4}$) changed from 30.0 (28.5) to 22.5 (20.0) mm, from 0.58 (0.56) to 0.41 (0.36), and from 0.81 (0.8) to 0.78 (0.76), respectively, by the use of bias correction. In both cases, the use of corrected reflectivity for rainfall estimation resulted in much better accuracy than did using raw reflectivity.

**4.2 Overlapping area method**

Figure 11 shows time series of the mean reflectivity differences between PSN and BSL in the overlapping area, and the number of samples used for calculation of $Z_H$ bias on 25 August 2014. Bias values ranged from −11.7 to −8.3 dB over the period analyzed. The bias was stable until 1440 LST, after which it fluctuated as the number of samples decreased. Figure 12 shows the same information for 8 September 2012. Bias values ranged from −4.66 to 0.22 dB, and lower bias values were occurred from 0300 LST to 0400 LST. The fluctuation also would be caused by the sudden change of microphysical characteristics of rainfall pass through the overlapping area for both radars. It would reduce the accuracy of $Z_H$ of BSL corrected by self-consistency. The radar rainfall estimation was done by using observed and corrected $Z_H$ as an input of Z-R relations.

Figure 13 shows a scatter plot of total accumulated radar rainfall amount for entire analyzed time period, calculated using $Z = 200 R^{1.6}$ and $Z=300R^{1.4}$ and gauge rainfall, for 25 August 2014 and 8 September 2012. The RMSE and NE of rainfall pairs for $Z = 200R^{1.6}$ ($Z=300R^{1.4}$) on 25 August 2014 were improved from 65.7 (66.1) to 29.7 (25.8) mm and from 0.79 (0.81) to 0.31 (0.28), respectively.  On 8 September 2012, RMSE and NE for $Z = 200R^{1.6}$ ($Z=300R^{1.4}$)

were improved from 30.0 (28.5) to 21.8 (19.1) mm and from 0.58 (0.56) to 0.40 (0.34), respectively, by the use of bias correction, while CC for $Z=200R^{1.6}$ was unchanged at 0.81 and that of $Z=300R^{1.4}$ were changed 0.8 to 0.79. Again, in both cases the use of corrected reflectivity for rainfall estimation was found to improve the accuracy compared with raw reflectivity.

## 4.3  Disdrometer method

Before using the disdrometer bias correction method to estimate rainfall rates, 10 min rain rates obtained directly from DSDs and from collocated gauges were compared. Figure 14 shows the time series of rain rate obtained by PARSIVEL and collocated gauges on 25 August 2014. Daily total rainfall amounts for PARSIVEL and the gauges were 129.4 and 116.0 mm, respectively.

The difference in the totals is only 13.4 mm, and the RMSE and CC between the 10 min time series were 0.52 mm $h^{-1}$ and 0.99, respectively. On 8 September 2012 (not shown), daily total rainfall amounts for PARSIVEL and the gauge were 54.4 and 55.0 mm, respectively. The difference between the total daily rainfall amounts was 0.7 mm and the RMSE and CC between the two 10 min series were 0.62 mm $h^{-1}$ and 0.96, respectively. It is concluded that DSDs were sufficiently reliable to use as a reference with which to calculate the radar bias.

Figure 15 shows time series of reflectivity obtained by radar and by PARSIVEL, and the radar bias, on 25 August 2014. The bias was more stable before 1200 LST than after 1500 LST.

PARSIVEL reflectivity fell to zero from 1230 to 1340 LST because the precipitation system moved away from the PARSIVEL site. The sudden change of rainfall would cause the unstable reflectivity difference from 1340 LST to 1500 LST. The threshold of reflectivity value observed from both PSN and PSRSIVEL should be considered for the comparison to get more reliable

$Z_H$ bias. The bias would be obtained more accurately when the reflectivity values observed from both instruments were higher than 15 dBZ in this event.   Because of this discontinuity, the bias can be considered to be reliable only until 1200 LST. The bias values ranged from -13.4 to -3.1

dB until 1200 LST.  Figure 16 shows time series of reflectivity obtained by radar and by

PARSIVEL, and the radar bias on 8 September 2012. On this occasion there was no reflectivity data from either PARSIVEL or radar until 0330 LST. The bias values were distributed from -

14.3 to 12.7dB.

Figure 17 shows a scatter plot of total accumulated radar rainfall amount for the entire time period, calculated using $Z = 200R^{1.6}$ and $Z=300R^{1.4}$ and gauge rainfall, on 25 August 2014 and

8 September 2012. The RMSE and NE of rainfall pairs for $Z = 200R^{1.6}$ ($Z=300R^{1.4}$) on 25

August 2014 were improved from 65.7 (66.1) mm to 42.0 (61.4) mm and from 0.79 (0.81) to 0.40 (0.53), respectively. On 8 September 2012, RMSE and NE for $Z = 200R^{1.6}$ ($Z=300R^{1.4}$) decreased from 30.1 (28.6) to 24.6 (23.9) mm, and from 0.58 (0.56) to 0.46 (0.44), respectively, while CC for $Z = 200R^{1.6}$ ($Z=300R^{1.4}$) decreased from 0.81 (0.8) to 0.65 (0.59). In both cases, using corrected rather than raw reflectivity for rainfall estimation improved accuracy as measured by RMSE and NE, but reduced accuracy as measured by CC.

**4.4 Discussion**

Figure 18 shows RMSE of total rainfall amount for entire time period obtained by gage and $Z=200R^{1.6}$ from each of the different bias correction methods on 25 August 2014 and 8 September 2012. Red, black, green, and blue bars show the RMSE obtained using the uncorrected, equidistance line, overlapping area, and disdrometer methods, respectively. The disdrometer method produced the lowest RMSE before 1200 LST and the highest RMSE after 1200 LST (Fig. 18(a)). This behavior can be attributed to the varying stability of the reflectivity calculated by PARSIVEL (Fig. 15). The overlapping method is more accurate than the equidistance line method for the entire time period, except at 1400 LST. All the bias correction methods performed better than the uncorrected method, except for the period during which DSDs were unavailable. On 8 September 2012, the RMSE of the overlapping area method was lower than that of the other methods for the entire period, except at 0500 and 0600 LST (Fig. 18(b)). The disdrometer method produced lower RMSE at 0600 LST, when DSDs were available, and the equidistance line method was more accurate at 0500 LST, when the sample number was high (Fig. 15). Comparing the RMSE between two events, the large fluctuation was occurred. It would be caused by the difference of total rainfall amount between two rainfall systems. The maximum total rainfall amount for both cases were around 250 mm for 25 August and 150 mm for 8 September 2012. Another reason of the fluctuation would be the difference of radar hardware calibration error for PSN between two events.

Considering the entire period covering both events, the overlapping area method showed the best performance, as measured by RMSE. The accuracy of radar rainfall estimates could be improved by combining the three approaches, using metrics such as DSD temporal stability and the number of samples available for the equidistance line method to select the best method for a particular situation. It is worth to noting that the result would be changed when the drop size distributions was fluctuated with height especially at the layer between radar beam and ground in case of disdrometer method.

**5 Conclusions**

Three methods for determining the reflectivity bias of single polarization radar using dual polarization radar reflectivity and disdrometer data were proposed and examined for two rainfall events caused by low pressure over the Korean Peninsula on 25 August 2014 and 8 September 2012. Single polarization radar reflectivity was underestimated by more than 12 dB and 7 dB during the August and September events, respectively. All three methods improved the accuracy of estimated rainfall, except during a period when DSDs were not observed (as the precipitation system did not pass over the disdrometer location).

The rainfall estimation using $Z = 200R^{1.6}$ and $Z=300R^{1.4}$ and gauge rainfall were examined for 25 August 2014 and 8 September 2012 to investigate the accuracy of each method. The RMSE, NE, and CC of rainfall pairs for $Z = 200R^{1.6}$ ($Z=300R^{1.4}$) on 25 August 2014 in case of using equidistance method were improved from 65.7 (66.1) to 32.6 (27.0) mm, from 0.79 (0.81) to 0.36 (0.31), and from 0.88 (0.87) to 0.89 (0.88), respectively. On 8 September 2012, the RMSE, NE, and CC for $Z = 200R^{1.6}$ ($Z=300R^{1.4}$) changed from 30.0 (28.5) to 22.5 (20.0) mm, from 0.58 (0.56) to 0.41 (0.36), and from 0.81 (0.8) to 0.78 (0.76), respectively.

The RMSE and NE of rainfall pairs for $Z = 200R^{1.6}$ ($Z=300R^{1.4}$) on 25 August 2014 in case of using overlapping method were improved from 65.7 (66.1) to 29.7 (25.8) mm and from 0.79 (0.81) to 0.31 (0.28), respectively. On 8 September 2012, RMSE and NE for $Z = 200R^{1.6}$ ($Z=300R^{1.4}$) were improved from 30.0 (28.5) to 21.8 (19.1) mm and from 0.58 (0.56) to 0.40 (0.34), respectively, by the use of bias correction, while CC for $Z=200R^{1.6}$ was unchanged at 0.81 and that of $Z=300R^{1.4}$ were changed 0.8 to 0.79.

The RMSE and NE of rainfall pairs for $Z = 200R^{1.6}$ ($Z=300R^{1.4}$) on 25 August 2014 in case of using disdrometer method were improved from 65.7 (66.1) mm to 42.0 (61.4) mm and from 0.79 (0.81) to 0.40 (0.53), respectively. On 8 September 2012, RMSE and NE for $Z = 200R^{1.6}$ ($Z=300R^{1.4}$) decreased from 30.1 (28.6) to 24.6 (23.9) mm, and from 0.58 (0.56) to 0.46 (0.44), respectively, while CC for $Z = 200 R^{1.6}$ ($Z=300R^{1.4}$) decreased from 0.81 (0.8) to 0.65 (0.59).

[revised manuscript text omitted]

(a)                              (b)

[Figure]

(c)                              (d)

(e)                              (f)

Figure 2. Time series of horizontal reflectivity (ZH) at 0.5 elevation angle observed from BSL

(a) 0400 LT, (c) 0500 LT, (e) 0600 LT on 8 September in 2012, (b) 1200 LT, (d) 1300 LT, (f)

1400 LT on 25 August in 2014.

(a)

[Figure]

(b)

[Figure]

Figure 3. Time series of 1 hour rainfall (bar) and daily accumulated (red line) measured from a gage which recorded highest daily rainfall within radar coverage at (a) North Changwon (ID

255) on 8 September in 2012 and (b) Geumjeong (ID 939) on 25 August in 2014.

[Figure]

2 Figure 4. Schematic diagram showing the method used to calculate the line of equidistance

3 between two radars. The effective radius was set to 100 km and the distance between radars is

4 76.9 km. The azimuthal angle from BSL to PSN is 147.6 degrees. The start and end azimuthal

5 angles are 79 (35) and 213 (261) degrees for BSL (PSN), respectively. The blue dashed line

6 shows the equidistance line.

[Figure]

Figure 5. Beam height of PSN (blue solid lines) and BSL (red dotted lines) at the equidistance line. EL1 to EL6 show the lowest, second, third, fourth, fifth, and sixth elevation angles, respectively.

[Figure]

Figure 6. Schematic diagram of the overlapping area for BSL and PSN. The east–west and
north–south distances between the two radars are 42 km and 64 km, respectively. The red (blue)
dotted circle shows the maximum range of BSL (PSN) and gray shaded area show 200 km by
km extracted from each radar coverage in the left panel.

[Figure]

Figure 7. Schematic diagram showing matching of the radar gate and the PARSIVEL

disdrometer. PARSIVEL is located ~9 km from the radar, at an azimuthal angle of 87 degrees.

The radar reflectivity was averaged over a 3 km × 3° domain centered at the PARSIVEL

location.

[Figure]

Figure 8. Time series of the average reflectivity difference between PSN and BSL at the equidistance line (blue circles) and the number of samples used in each calculation (black squares) on 25 August in 2014.

[Figure]

Figure 9. As for Fig. 8 but for 8 September 2012.

[Figure]

Figure 10. Scatter plot of total accumulated rainfall for analyzed time period calculated by gage and radar using (a and b) $Z = 200\ R^{1.6}$ and (c and d) $Z = 300\ R^{1.4}$ for 25 August 2014 and 8 September 2012, respectively. Blue circles show the rainfall pairs obtained using raw reflectivity and red circles show those obtained using reflectivity corrected with the equidistance line method.

[Figure]

Figure 11. Time series of the average reflectivity difference between PSN and BSL at the overlapping area (blue circles) and the number of samples used in each calculation (black squares) on 25 August in 2014.

[Figure]

Figure 12. Time series of the average reflectivity difference between PSN and BSL at the overlapping area (blue circles) and the number of samples used in each calculation (black squares) on 8 September in 2012.

(a)                                              (b)

[Figure]

(c)                                              (d)

Figure 13. As for Fig. 10 but for the overlapping area method.

[Figure]

Figure 14. Time series of 10 min rainfall amount as obtained by PARSIVEL (red circles) and collocated gauges (blue circles).

[Figure]

Figure 15. Time series of reflectivity obtained by  PARSIVEL (red circles), and the radar bias (blue circles) on 25 August 2014.

[Figure]

Figure 16. As for Fig. 15 but for 8 September 2012.

(a)                                                                 (b)

[Figure]

(c)                                                                 (d)

Figure 17. As for Fig. 10 but for the disdrometer method.

(a) (b)

[Figure]

Figure 18. Accumulated rainfall RMSE calculated from radar and gage for different bias correction methods on (a) 25 August 2014 and (b) 8 September 2012. The bars with different colors show results obtained using the raw data, equidistance line method, overlapping area method, and disdrometer method, respectively.